# Integrative genomic analyses identify candidate causal genes for calcific aortic valve stenosis involving tissue-specific regulation

Sébastien Thériault [1,2] ✉, Zhonglin Li [1], Erik Abner [3], Jian'an Luan [4], Hasanga D. Manikpurage [1], Ursula Houessou [1], Pardis Zamani[1], Mewen Briend [1], Estonian Biobank Research Team*, Dominique K. Boudreau [1], Nathalie Gaudreault [1], Lily Frenette[1], Déborah Argaud[1], Manel Dahmene[1], François Dagenais[1,5], Marie-Annick Clavel [1,6], Philippe Pibarot[1,6], Benoit J. Arsenault [1,6], S. Matthijs Boekholdt[7], Nicholas J. Wareham [4], Tõnu Esko[3], Patrick Mathieu [1,5] & Yohan Bossé [1,8]

There is currently no medical therapy to prevent calcific aortic valve stenosis (CAVS). Multi-omics approaches could lead to the identification of novel molecular targets. Here, we perform a genome-wide association study (GWAS) meta-analysis including 14,819 cases among 941,863 participants of European ancestry. We report 32 genomic loci, among which 20 are novel. RNA sequencing of 500 human aortic valves highlights an enrichment in expression regulation at these loci and prioritizes candidate causal genes. Homozygous genotype for a risk variant near *TWIST1*, a gene involved in endothelial-mesenchymal transition, has a profound impact on aortic valve transcriptomics. We identify five genes outside of GWAS loci by combining a transcriptome-wide association study, colocalization, and Mendelian randomization analyses. Using cross-phenotype and phenome-wide approaches, we highlight the role of circulating lipoproteins, blood pressure and inflammation in the disease process. Our findings pave the way for the development of novel therapies for CAVS.

Calcific aortic valve stenosis (CAVS) is the most frequent valvular heart disease, affecting up to 2% of the population aged 65 years and older[1,2]. Progressive fibro-calcific remodeling of aortic valve leaflets leads to the obstruction of blood flow from the left ventricle to the aorta, causing left ventricular remodeling, heart failure, and death[3,4]. Aortic valve replacement is currently the only effective treatment and there is no pharmacological therapy to prevent CAVS or slow its progression.

In the last decade, several genetic variants associated with CAVS have been identified through genome-wide association studies (GWAS), improving our understanding of the disease pathophysiology. Variants in *LPA*, associated with lipoprotein (a) levels, and *PALMD*,

[1]Institut universitaire de cardiologie et de pneumologie de Québec-Université Laval, Quebec City, QC, Canada. [2]Department of Molecular Biology, Medical Biochemistry and Pathology, Université Laval, Quebec City, QC, Canada. [3]Estonian Genome Center, Institute of Genomics, University of Tartu, Tartu, Estonia. [4]MRC Epidemiology Unit, Institute of Metabolic Science, University of Cambridge, Cambridge, United Kingdom. [5]Department of Surgery, Université Laval, Quebec City, QC, Canada. [6]Department of Medicine, Université Laval, Quebec City, QC, Canada. [7]Department of Cardiology, Amsterdam University Medical Centers, University of Amsterdam, Amsterdam, The Netherlands. [8]Department of Molecular Medicine, Université Laval, Quebec City, QC, Canada. *A list of authors and their affiliations appears at the end of the paper. ✉e-mail: sebastien.theriault@criucpq.ulaval.ca

a regulator of actin polymerization[5,6], remain the most strongly associated with CAVS[7–9]. Large-scale transcriptomic analyses in human aortic valves have confirmed the specific role of *PALMD* in CAVS[8,10], and led to the identification of *NAV1* as another contributing gene[9]. Variants located in *TEX41*, *IL6*, *ALPL* and *FADS1/2* have also shown significant association with CAVS in previous GWAS[9,11,12]. More recently, two large GWAS including 14,451 and 13,765 cases, respectively, led to the identification of 20 additional loci, some of which are associated with blood lipids (*CELSR2-SORT1*) or obesity (*FTO*)[13,14].

In this work, we perform a GWAS meta-analysis for CAVS in 14,819 cases among 941,863 participants and generate extensive transcriptomic data from 500 human aortic valves to identify novel key molecular drivers. Our objectives were to (1) identify novel genetic loci and candidate causal genes that impact gene expression in the aortic valve and (2) characterize the relationship between the genetic architecture of CAVS and cardiovascular traits.

## Results

### Identification of CAVS genomic loci
A GWAS meta-analysis for CAVS was performed in 14,819 cases (61% men) and 927,044 controls of European ancestry from six cohorts (Supplementary Fig. 1, Supplementary Fig. 2, and Supplementary Data 1). In total, 17,130,226 variants were included in the meta-analysis, 7,159,971 of which were available in all six cohorts. The genomic inflation factor was 1.099 and the LD score regression intercept was 1.020. Thirty-two loci reached genome-wide significance ($P < 5 \times 10^{-8}$), including 20 novel loci (Table 1, Fig. 1 and Supplementary Data 2). Among the novel associations, variants located in *LPL* and *LDLR*, two major regulators of circulating blood lipids, were identified. The 32 lead variants at the genome-wide associated loci had a concordant direction of effect in the recent meta-analysis by Chen et al.[14]. Thirty showed nominal association with CAVS ($P < 0.05$), of which 29 remained significant when using a threshold of false discovery rate <5%. In the deCODE cohort ($n = 2457$ CAVS cases and 349,342 controls)[11], 28 out of the 32 lead variants had the same direction of effect (Supplementary Data 3). Variants at seven other loci previously reported in European populations were nominally associated ($P < 0.05$) with a concordant direction of effect, including six that remained significant when using a threshold of false discovery rate <5% (Supplementary Data 4).

### Variant annotation and prioritization
Among variants in linkage disequilibrium with independent significant SNPs, there was a significant enrichment for intronic and exonic functional annotations (Supplementary Data 5). Among the lead SNPs, two were missense variants located in *STARD9* (rs28744617) and *ASCC2* (rs61736786). Missense variants in *ALDH1A2* (rs4646626), *MUC4* (rs2246901), and *CDAN1* (rs4265781) were in high LD ($r^2 \geq 0.8$) with a lead SNP. The missense variants in *ASCC2*, *ALDH1A2* and *MUC4* had Combined Annotation Dependent Depletion (CADD) scores of 14.52, 17.37, and 21.50, respectively, ranking them in the 5% most deleterious variants in the genome. A nonsense variant in *LPL* previously associated with circulating triglyceride levels, rs328 (S447X), was in high LD ($r^2 = 0.958$) with the lead variant at this locus (Supplementary Data 6). The 95% credible set, determined using a Bayesian approach implemented in CAVIAR, included five or less variants for five loci (nearest genes: *PALMD*, *PRRX1*, *TMEM44*, *LPA*, *PDE3A* and *HMGA2*). The complete sets for each locus are reported in Supplementary Data 7. Conditional analyses led to the identification of independent genome-wide significant signals at the *LPA* ($n = 3$) and *IL6* ($n = 2$) loci (Supplementary Data 8).

### Gene mapping
We mapped the protein-coding genes located within 10 kilobases (kb) of a genome-wide significant SNP and the genes located on either side

for lead intergenic SNPs, identifying a list of 79 nearby genes (Supplementary Data 2). A gene-based analysis using MAGMA identified 158 genes (117 additional genes) at a false discovery rate threshold <5%, for a total of 196 prioritized genes (Supplementary Data 9).

### Expression in human aortic valves
To identify the genes most likely to be involved in CAVS, we completed extensive transcriptomic profiling of the most relevant tissue, the aortic valve. We performed RNA-sequencing on 500 human aortic valve samples of various disease stages (81.6% with CAVS) and valve morphology (44% bicuspid) from patients recruited in our institution to create the QUEBEC-CAVS-RNA dataset (Supplementary Data 10). First, the expression level of the 196 genes prioritized by positional mapping or MAGMA was evaluated. Among the 187 genes with available gene expression quantification in aortic valves, 22 had a median expression level above the 90th percentile of all the protein-coding genes, including seven genes located nearby the meta-analysis lead SNPs: *RPLP2*, *CCND1*, *ACTR2*, *TALDO1*, *PDGFRA*, *PRRX1*, and *UQCR10* (Supplementary Data 11).

To identify genes for which there is specific expression in the aortic valve, we compared gene expression with 43 GTEx tissues by calculating expression specificity scores (ESS). Among the genes of interest, nine had an ESS > 0.1 for the aortic valve, which corresponded approximately to the top 5% of the empirical distribution for all the protein-coding genes in the aortic valve (Supplementary Data 12). Five of them were located nearby the meta-analysis lead SNPs: *CHST6*, *CCND1*, *ALDH1A2*, *ADAMTS7* and *PRRX1*. The maximum ESS was observed in the aortic valve for all five genes except *PRRX1*, for which there was a slightly higher ESS in fibroblasts in GTEx. Other genes with a high ESS in a relevant tissue included *HMGA2* (ESS = 0.80 in fibroblasts), *NKX2-5* (ESS = 0.42 in atrial appendage), *PDE3A* (ESS = 0.14 in tibial artery) and *PDGFRA* (ESS = 0.13 in fibroblasts) (Supplementary Data 12 and Supplementary Fig. 3).

### Expression quantitative trait loci
Genome-wide expression quantitative trait loci (eQTL) analyses in 484 human aortic valves identified 4,671,347 significant SNP-gene pairs. Among the 32 meta-analysis lead SNPs, 48 significant SNP-gene pairs were identified in the aortic valve, located in 20 loci (Supplementary Data 13). For these 32 lead SNPs, there was a significantly higher proportion of significant SNP-gene pairs (48/915, 5.25%) compared to all the SNP-gene pairs tested (4,671,347/182,925,823, 2.55%, $P = 4.25 \times 10^{-7}$). A Wilcoxon rank sum test also confirmed an enrichment towards stronger associations for these 915 SNP-gene pairs ($P = 1.71 \times 10^{-7}$) (Supplementary Fig. 4).

In 43 tissues from GTEx, significant eQTLs were found for 25 loci (Supplementary Data 14). The number of significant SNP-gene pairs ranged from one (brain anterior cingulate cortex) to 37 (tibial nerve). Taken together, the most significant eQTL was in the aortic valve for 12 loci, including four genes for which there was no reported eQTL in the GTEx tissues: rs6702619-*PALMD* ($P = 7.1 \times 10^{-119}$), rs7804522-*TWIST1* ($P = 5.7 \times 10^{-22}$), rs1965668-*NKX2-5* ($P = 7.9 \times 10^{-12}$) and rs72854462-*TEX41* ($P = 7.8 \times 10^{-6}$). For 16 other loci, the most significant eQTL was found in a GTEx tissue. However, these were distributed among ten different tissues, which each had between one and three of the most significant SNP-gene pairs for a given locus (Supplementary Fig. 5). Among these, associations include rs1706003-*ATP13A3* in the left ventricle ($P = 1.7 \times 10^{-26}$), rs771264-*RNF144A* in fibroblasts ($P = 1.9 \times 10^{-12}$) and rs11330858-*MECOM* in the aorta ($P = 7.5 \times 10^{-5}$).

### Transcriptome-wide association study, colocalization and Mendelian randomization
A transcriptome-wide association study (TWAS) was performed to identify genes for which genetically predicted expression in the aortic valve is associated with CAVS. Thirty-five genes reached statistical

**Table 1 | Genome-wide significant loci for calcific aortic valve stenosis from the meta-analysis**

| Locus | rsID | Chr | Pos | RA/NRA | RAF | OR (95% CI) | P | I² (%) | P_Het | Gene | Novel |
|---|---|---|---|---|---|---|---|---|---|---|---|
| 1 | rs2077522 | 1 | 87917746 | C/T | 0.592 | 1.08 (1.06–1.11) | 3.1E−10 | 0.0 | 0.60 | *LMO4* | Yes |
| 2 | rs6702619 | 1 | 100046246 | G/T | 0.496 | 1.18 (1.15–1.21) | 3.9E−40 | 0.0 | 0.51 | *PALMD* | No |
| 3 | rs1016819 | 1 | 170645774 | G/C | 0.341 | 1.11 (1.08–1.14) | 1.2E−14 | 0.0 | 0.81 | **PRRX1** | No |
| 4 | rs682112 | 1 | 201746768 | A/G | 0.398 | 1.11 (1.08–1.14) | 3.2E−14 | 49.9 | 0.09 | **NAV1** | No |
| 5 | rs771264 | 2 | 7129124 | C/T | 0.496 | 1.07 (1.04–1.10) | 5.0E−08 | 69.1 | 0.0064 | **RNF144A** | Yes |
| 6 | rs62139061 | 2 | 65498805 | C/T | 0.300 | 1.09 (1.06–1.12) | 4.0E−10 | 24.6 | 0.25 | **ACTR2** | No |
| 7 | rs72854462 | 2 | 145720139 | A/G | 0.735 | 1.15 (1.12–1.19) | 1.7E−22 | 16.4 | 0.31 | *ZEB2*ᵃ | No |
| 8 | rs111825950 | 3 | 153748291 | T/TTTCA | 0.197 | 1.11 (1.08–1.15) | 1.1E−10 | 0.0 | 0.49 | **ARHGEF26** | No |
| 9 | rs11330858 | 3 | 169202578 | A/AT | 0.489 | 1.08 (1.05–1.11) | 1.3E−09 | 0.0 | 0.43 | **MECOM** | No |
| 10 | rs1706003 | 3 | 194299967 | G/T | 0.542 | 1.09 (1.06–1.12) | 3.1E−11 | 0.0 | 0.69 | *TMEM44* | Yes |
| 11 | rs3103933 | 3 | 195485440 | A/G | 0.305 | 1.12 (1.08–1.16) | 6.1E−09 | 33.6 | 0.20 | **MUC4** | Yes |
| 12 | rs147558377 | 4 | 55113623 | G/C | 0.149 | 1.10 (1.07–1.14) | 2.4E−08 | 0.0 | 0.79 | **PDGFRA** | Yes |
| 13 | rs182432302 | 5 | 45560981 | T/C | 0.970 | 1.33 (1.20–1.47) | 2.3E−08 | 0.0 | 0.98 | *HCN1* | Yes |
| 14 | rs1965668 | 5 | 172653401 | G/C | 0.687 | 1.08 (1.05–1.11) | 1.6E−08 | 0.0 | 0.65 | **NKX2-5** | Yes |
| 15 | rs10455872 | 6 | 161010118 | G/A | 0.063 | 1.53 (1.45–1.61) | 3.0E−61 | 52.3 | 0.063 | **LPA** | No |
| 16 | rs7804522 | 7 | 19458058 | C/G | 0.421 | 1.07 (1.05–1.10) | 1.9E−08 | 28.4 | 0.22 | *FERD3L* | Yes |
| 17 | rs1800797 | 7 | 22766221 | A/G | 0.480 | 1.12 (1.09–1.15) | 1.3E−19 | 0.0 | 0.96 | **IL6** | No |
| 18 | rs2286427 | 7 | 76024520 | T/C | 0.162 | 1.10 (1.06–1.14) | 4.6E−08 | 0.0 | 0.91 | **SSC4D** | Yes |
| 19 | rs11570891 | 8 | 19822810 | C/T | 0.894 | 1.20 (1.13–1.28) | 7.2E−09 | 0.0 | 0.62 | **LPL** | Yes |
| 20 | rs17810852 | 9 | 136365075 | G/A | 0.458 | 1.08 (1.05–1.10) | 1.1E−08 | 0.0 | 0.43 | *MYMK* | Yes |
| 21 | rs187229435 | 11 | 805953 | C/G | 0.021 | 1.29 (1.19–1.40) | 1.6E−09 | 56.3 | 0.043 | *PIDD1* | Yes |
| 22 | rs174551 | 11 | 61573684 | T/C | 0.620 | 1.09 (1.06–1.12) | 3.1E−10 | 0.6 | 0.40 | **FADS1** | No |
| 23 | rs12270146 | 11 | 69163441 | G/A | 0.322 | 1.09 (1.06–1.12) | 8.3E−10 | 0.0 | 0.89 | **MYEOV** | Yes |
| 24 | rs10770612 | 12 | 20230639 | A/G | 0.779 | 1.12 (1.08–1.15) | 6.6E−13 | 40.9 | 0.13 | *PDE3A* | Yes |
| 25 | rs17766960 | 12 | 66407417 | G/T | 0.816 | 1.10 (1.07–1.14) | 1.6E−09 | 0.0 | 0.61 | *HMGA2* | Yes |
| 26 | rs12429277 | 13 | 37445944 | A/C | 0.247 | 1.08 (1.05–1.12) | 2.3E−08 | 0.0 | 0.50 | **SMAD9** | Yes |
| 27 | rs28744617 | 15 | 42981022 | A/G | 0.763 | 1.09 (1.06–1.12) | 2.2E−08 | 0.0 | 0.90 | **STARD9** | No |
| 28 | rs4646642 | 15 | 58246916 | A/G | 0.446 | 1.08 (1.05–1.10) | 4.2E−09 | 0.0 | 0.72 | **ALDH1A2** | Yes |
| 29 | rs2869553 | 15 | 79012613 | G/T | 0.240 | 1.09 (1.05–1.12) | 2.9E−08 | 76.0 | 0.00086 | *ADAMTS7* | Yes |
| 30 | rs150429885 | 16 | 75498252 | CT/C | 0.613 | 1.08 (1.06–1.11) | 1.9E−09 | 0.0 | 0.97 | **TMEM170A** | No |
| 31 | rs6511720 | 19 | 11202306 | G/T | 0.892 | 1.12 (1.08–1.17) | 2.5E−08 | 0.0 | 0.89 | *LDLR* | Yes |
| 32 | rs61736786 | 22 | 30189642 | T/C | 0.043 | 1.19 (1.12–1.26) | 3.5E−08 | 35.5 | 0.17 | **ASCC2** | Yes |

The association of each variant with calcific aortic valve stenosis was obtained from an inverse-variance weighted fixed-effect meta-analysis combining the effect per allele in the cohorts with available data.

Genes in bold are significant in MAGMA gene-based analysis (*P* < 0.00039 corresponding to false discovery rate <5%).

*Pos* position according to GRCh37, *RA* risk allele, *NRA* non-risk allele, *RAF* risk allele frequency, *I²* heterogeneity statistic, *P*_Het heterogeneity *p*-value (Cochran's Q-test), *Gene* Nearest protein-coding gene.

ᵃLocated in lncRNA *TEX41*

significance at a false discovery rate <5%, including *PALMD* and *NAV1*, identified in a previous TWAS[9] (Supplementary Data 15). To obtain further evidence on the potential causal role of these genes in CAVS, colocalization and Mendelian randomization (MR) analyses were performed. After removing genes with a low probability for colocalization of the GWAS and eQTL signals (PP4 < 0.75, $n = 20$), with no MR instrument ($n = 3$) or with high heterogeneity of the instrument ($n = 2$), ten genes remained (*PALMD*, *NAV1*, *PRRX1*, *ATP13A3*, *BCL10*, *TWIST1*, *RAD9A*, *NRBP1*, *FES* and *AFAP1*). All genes showed a significant association in MR with the inverse-variance weighted and weighted median approaches ($P_{IVW}$ and $P_{WM}$ adjusted for false discovery rate <5%), suggesting a causal association between gene expression in the aortic valve and CAVS. There was no evidence of pleiotropy with the Egger intercept test ($P_{intercept} > 0.05$ for all genes). Among the novel genes identified, three were located at a genome-wide significant locus: *PRRX1*, *ATP13A3* and *TWIST1*, for which gene expression in the aortic valve was positively associated with CAVS ($P_{IVW} = 4.5 \times 10^{-17}$, $9.4 \times 10^{-9}$, and $5.7 \times 10^{-4}$, respectively) (Fig. 2). The lead SNPs in the meta-analysis at these three loci were strong eQTLs in the aortic valve (all

$P_{eQTL} < 1 \times 10^{-15}$). Five other genes were identified; the association between genetically predicted expression and CAVS was positive for three: *BCL10*, *RAD9A* and *NRBP1* ($P_{IVW} = 0.0066$, $5.6 \times 10^{-5}$, and $9.4 \times 10^{-4}$, respectively) and negative for two: *FES* and *AFAP1* ($P_{IVW} = 7.9 \times 10^{-4}$ and $5.5 \times 10^{-4}$, respectively) (Supplementary Figs. 6 and 7).

To evaluate a potential impact of valve morphology on gene expression regulation for these genes, we calculated eQTL separately for tricuspid and bicuspid aortic valves. For all 10 genes identified by the transcriptomic analyses, the effect of the lead variant on gene expression was similar in tricuspid and bicuspid valves (all $P < 0.05$ for heterogeneity) (Supplementary Data 16).

## Differential expression according to genotype at the *TWIST1* locus

Considering the potential causal relationship between genetically determined expression of *TWIST1* in the aortic valve and CAVS risk, as well as the fact that the lead GWAS SNP, rs7804522, is only associated with gene expression in the aortic valve (valve-specific eQTL), we

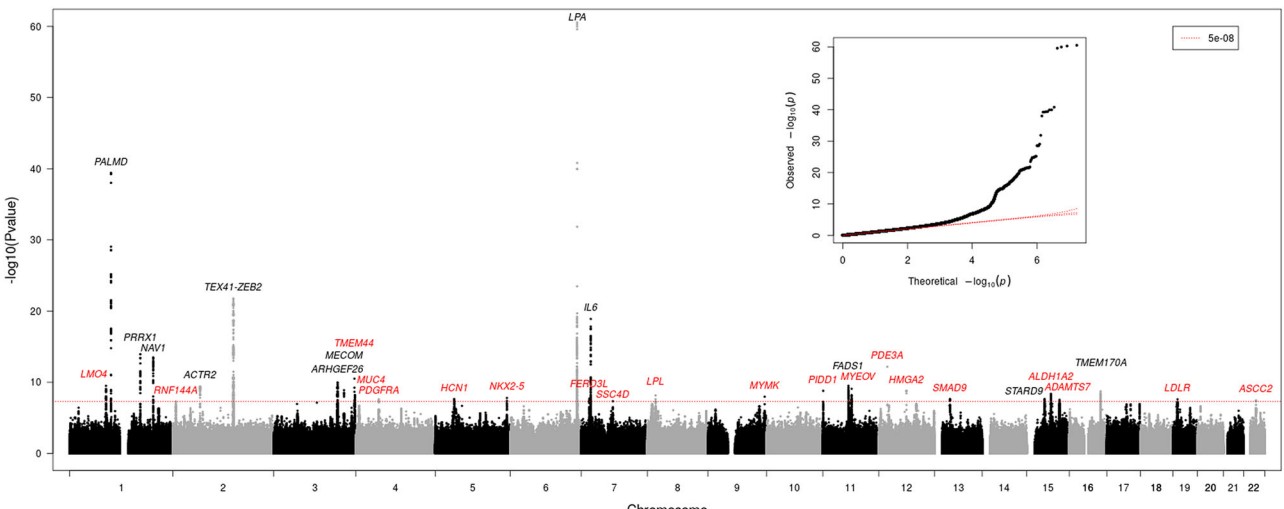

**Fig. 1 | Manhattan plot of the genome-wide association studies meta-analysis for CAVS.** The nearest gene at each genome-wide significant locus is indicated, in black for known loci and in red for novel loci. The association of each variant with calcific aortic valve stenosis was obtained from an inverse-variance weighted fixed-effect meta-analysis combining the effect per allele in the cohorts with available data. A p-value below $5 \times 10^{-8}$ was considered significant (genome-wide threshold). The genomic inflation factor was 1.099 and the LD score regression intercept was 1.020. The quantile-quantile plot (inset) illustrates the distribution of p-values for each variant tested.

explored further the impact of this variant on the transcriptomic profile in the aortic valve. After selecting individuals with severe aortic stenosis, we compared gene expression between the 66 individuals homozygous for the risk allele (rs7804522-C) and the 138 individuals homozygous for the other allele. We found 509 differentially expressed genes at a false discovery rate <5%, indicating a profound effect of this genotype on transcriptome-wide gene expression (Fig. 3 and Supplementary Data 17). For comparison, a differential expression analysis performed using the same method for the four other genome-wide significant loci corroborated by transcriptomic analyses in the aortic valve (*PALMD*, *NAV1*, *PRRX1* and *ATP13A3*) showed only five or less differentially expressed genes between individuals homozygous for the risk allele and those homozygous for the other allele (Supplementary Data 18). At the *TWIST1* locus, a total of 148 genes were up-regulated while 361 genes were down-regulated in the group homozygous for the risk allele (Fig. 3). The dysregulated genes were enriched for several metabolic pathways including neutrophil degranulation, cell cycle, cell division and apoptosis (Fig. 3 and Supplementary Data 19).

### Prioritization of causal genes
To prioritize potential causal genes, we combined the evidence from ten different features, including lead variants location and annotation, association at the gene level, expression in the aortic valve, colocalization, TWAS and MR. Taken together, 17 genes located at genome-wide significant loci and eight genes identified by TWAS had four or more features that suggested their implication in CAVS (Fig. 4). Among the genes located at genome-wide significant loci, *PRRX1*, *NAV1*, *ALDH1A2* and *MUC4* had seven or more features while *ACTR2*, *SMAD9*, *PALMD*, *ATP13A3*, *FADS2* and *TWIST1* had four or more features including significant associations in the TWAS and MR analyses. Out of the 25 prioritized genes, seven encoded druggable human proteins (*MUC4*, *ALDH1A2*, *NRBP1*, *FES*, *PDGFRA*, *LPL* and *NPC1*). Existing drugs interacting with these genes according to the drug-gene interaction database are reported in Supplementary Data 20.

### Pathway enrichment
A pathway analysis was performed with the Metascape tool by including the genes identified using MAGMA as well as the 35 genes significant in the TWAS analysis. Among the top significantly enriched terms, we identified regulation of interleukin-6 production, embryonic

development, regulation of osteoblast differentiation, response to growth factor and plasma lipoprotein assembly, remodeling, and clearance (Supplementary Fig. 8 and Supplementary Data 21).

### Cross-phenotype analyses
We used the interactive cross-phenotype analysis of GWAS (iCPAG) database to identify other phenotypes that share genetic associations. After adjustment for multiple testing, there were 35 significantly associated phenotypes ($P_{Bonferroni} < 0.05$), including plasma lipids (lipoprotein (a), LDL-cholesterol, triglycerides, apolipoprotein B, total cholesterol, lipoprotein-associated phospholipase A2), blood pressure traits (pulse, diastolic and systolic blood pressure), other cardiovascular diseases (coronary artery disease, peripheral arterial disease, carotid atherosclerosis, metabolic syndrome), but also aortic root size, coronary artery calcification, bone mineral density, resting heart rate and leukocyte count (Supplementary Data 22).

The association of the lead GWAS and TWAS variants with 44 other cardiovascular traits was then evaluated in UK Biobank (Fig. 5). The CAVS risk allele for several variants was positively associated with circulating lipids such as LDL-cholesterol and apolipoprotein B or blood pressure traits. Notably, 14 variants had a significant association with pulse pressure while 11 were associated with coronary artery disease with a direction of effect concordant with CAVS.

Accordingly, there was a significant positive genetic correlation between CAVS and lipids, blood pressure traits and coronary artery disease. A positive genetic correlation was also observed for abdominal aorta calcification, ischemic stroke, peripheral artery disease, body-mass index, diabetes and C-reactive protein (Fig. 6 and Supplementary Data 23).

## Discussion
We performed a GWAS meta-analysis including 14,819 CAVS cases and identified 32 genome-wide significant loci, 20 of which had not been reported before, including 18 with supportive evidence from another study. Leveraging transcriptomic data obtained from RNA sequencing of 500 human aortic valves, we identified novel candidate causal genes, some of which could constitute therapeutic targets. We report several significant eQTL at genomic loci, many of which are specific to the aortic valve, implicating tissue-specific regulatory mechanisms. We also show robust evidence of genetic relationship between CAVS and

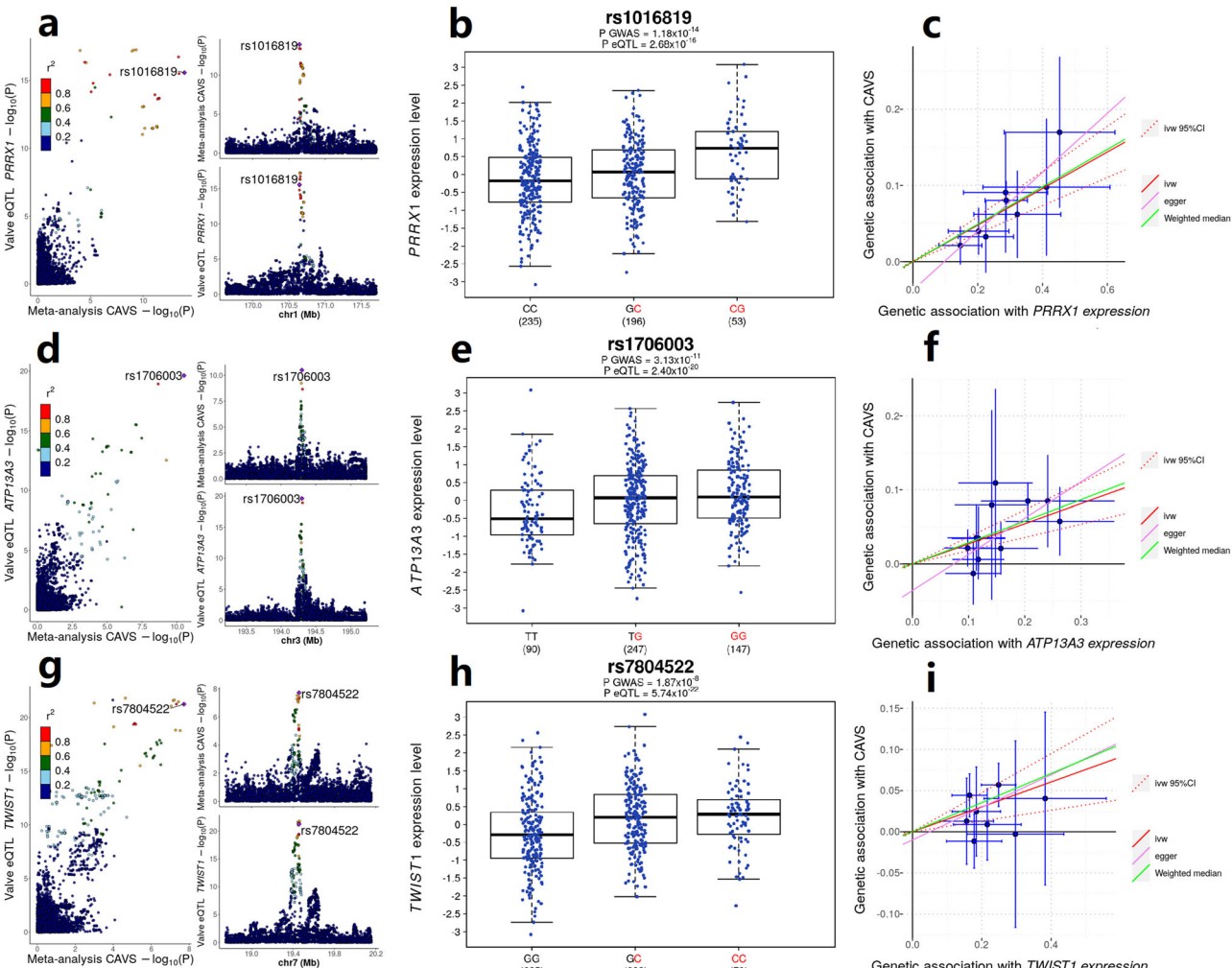

**Fig. 2 | Transcriptome-wide association study in human aortic valve and Mendelian randomization identify novel candidate causal genes at genome-wide significant loci. a**, **d**, **g** LocusCompare plots at the *PRRX1*, *ATP13A3*, and *TWIST1* loci. *P* for calcific aortic valve stenosis was obtained from the inverse-variance weighted fixed-effect GWAS meta-analysis. *P* for valve eQTL was obtained from the nominal association between genotype and normalized gene expression. **b**, **e**, **h** Boxplots showing normalized gene expression in human aortic valves according to the genotype at the lead SNP at the *PRRX1*, *ATP13A3*, and *TWIST1* loci. The center mark in the box represents the median, the bounds of the box represent the 25th and 75th percentiles and the whiskers are the most extreme data point, which is no more than 1.5 times the interquartile range. *P* GWAS was obtained from

the inverse-variance weighted fixed-effect GWAS meta-analysis. *P* eQTL was obtained from the nominal association between genotype and normalized gene expression. The allele in red is the risk allele. **c**, **f**, **i** Scatterplot representing the effect of each SNP selected in the instrument for the Mendelian randomization (MR) analysis on gene expression in human aortic valves (*n* = 484) and risk of calcific aortic valve stenosis (*n* = 14,819 cases and 927,044 controls) at the *PRRX1*, *ATP13A3*, and *TWIST1* loci. Data are presented as the effect and 95% confidence interval (+/−1.96*standard error). Red line: inverse-variant weighted (IVW) MR; Dotted red lines: 95% confidence interval for IVW MR; Green line: Weighted median MR; Pink line: Egger MR.

several traits, including circulating lipids, blood pressure, atherosclerotic cardiovascular diseases and inflammation.

Combining the largest GWAS meta-analysis to date with robust eQTL for the aortic valve generated from the most comprehensive resource so far led to the identification of target genes residing within genomic risk loci and revealed novel risk loci. A TWAS followed by Bayesian colocalization and MR identified eight genes for which genetically determined expression in the aortic valve is robustly associated with CAVS. Three of them were located at genome-wide significant loci: *PRRX1*, *ATP13A3* and *TWIST1*. *PRRX1* and *TWIST1* code for transcription factors involved in epithelial-mesenchymal transition, acting as potent mesenchymal promoters[15]. They were both identified as key members of fibroblast transcriptional regulatory networks and involved in a positive feedback loop that leads to fibroblast activation[16,17]. *PRRX1*, paired related homeobox 1, codes for a DNA-associated protein located in the nucleus that acts as a transcription co-activator. It is involved in embryonic development, where it was

shown to impact heart looping and laterality in animal models[18]. *PRRX1* has been shown to orchestrate the transition of stromal fibroblasts to myofibroblasts via TGF-β signaling[19], which is a known mechanism taking place in valve interstitial cells in CAVS[20]. Here, we show that *PRRX1* is highly expressed in human aortic valves, that the lead SNP at this locus is a strong eQTL specific to this tissue, and report a potential causal role of local *PRRX1* expression in CAVS. *TWIST1*, coding for twist family bHLH transcription factor 1, plays an important role in endothelial-mesenchymal transition and embryonic development for several organs, including the heart[21]. *TWIST1* is highly expressed in endocardial cushions during early development and is then down-regulated for heart valve remodeling[22]. Persistent expression in valves has been shown to increase cell proliferation, dysregulation of fibrillary collagen and lead to enlarged hypercellular valve leaflets in a mouse model. In the same study, high *TWIST1* expression was observed in human diseased aortic valves[23]. In accordance with these previous observations, we identified a significant positive association

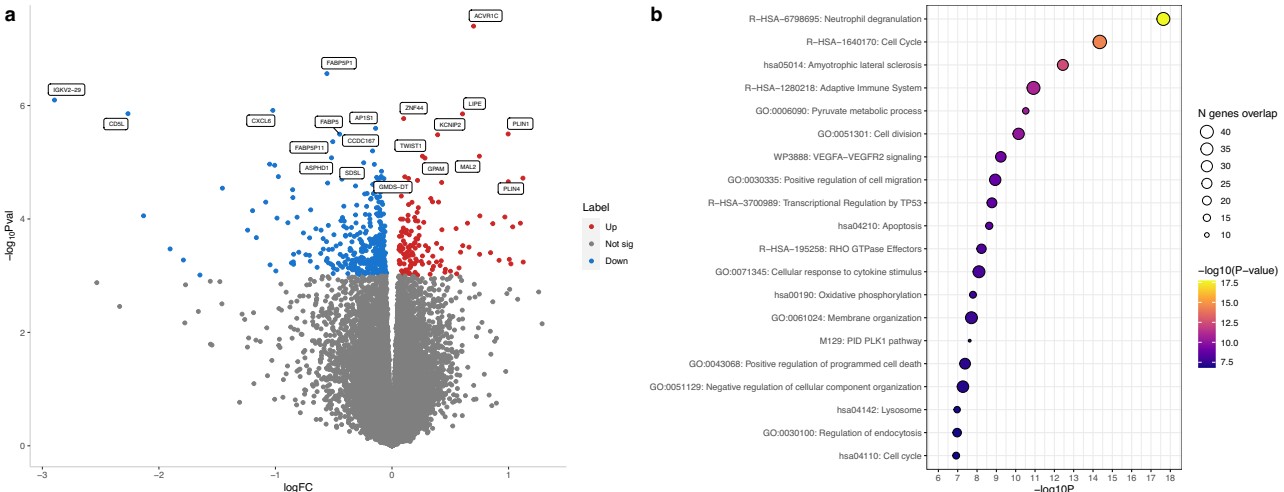

**Fig. 3 | Impact of homozygous risk genotype near *TWIST1* on aortic valve gene expression. a** Volcano plot representing the differentially expressed genes between individuals homozygous for rs7804522-C and individuals homozygous for rs7804522-G. Red points represent up-regulated genes (*n* = 148). Blue points represent down-regulated genes (*n* = 361). Statistical significance was set at *P* ≤ 0.001 corresponding to false-discovery rate <5% in the differential expression analysis. The top 10 up and down-regulated genes are labeled. **b** Lead independent enriched terms for the dysregulated genes. The statistical significance of the association for each term was obtained from a hypergeometric test and is illustrated by the color (*p*-value). The number of overlapping genes is illustrated by the size of the bubble.

between genetically determined expression of *TWIST1* in human aortic valves and CAVS. Moreover, the lead variant at this locus (rs7804522, minor allele frequency of 0.42) was a strong eQTL specific to the aortic valve with no reported association in GTEx, despite high expression levels in other tissues. Such remarkable eQTL specificity for the aortic valve was also observed at the *PALMD* locus, as reported previously[8]. We show that the aortic valves of individuals with severe CAVS carrying two risk alleles associated with increased *TWIST1* expression have a distinct transcriptomic profile compatible with increased cell proliferation, compared to individuals with CAVS of comparable hemodynamic severity carrying no risk allele. *ATP13A3* codes for a P-type ATPase involved in polyamine transport[24]. It is highly expressed in heart and vascular tissues. Protein-truncating variants in this gene have previously been associated with pulmonary artery hypertension. Functional studies have linked the loss of ATP13A3 to inhibition of proliferation and stimulation of apoptosis in endothelial cells[25]. In our study, the risk allele at the lead SNP was associated with an increase in *ATP13A3* expression in the aortic valve. The same variant was also significantly associated with lower diastolic blood pressure and higher QT interval in previous GWAS[26,27].

Five other genes were identified as candidate causal genes based on TWAS, colocalization and MR: *BCL10*, *RAD9A* and *NRBP1* had a positive association, while *FES* and *AFAP1* had a negative association with CAVS. *BCL10* is an immune signaling adaptor mostly expressed in lymphocytes. It has a key role in lymphocyte activation through the nuclear factor-kappa B pathway[28]. Homozygous loss-of-function mutations in this gene cause severe combined immunodeficiency[29]. Higher expression of *BCL10* in the aortic valve could contribute to the immune-related inflammation observed in CAVS[20]. *RAD9A* encodes a checkpoint protein required for cell cycle arrest and DNA damage repair. DNA binding to RAD9A was shown to have a critical role in maintaining cell viability and checkpoint activation in the presence of oxidative stress[30]. To our knowledge, this gene has not been associated with cardiovascular traits before. *NRBP1*, nuclear receptor binding protein 1, is involved in endoplasmic reticulum to Golgi vesicle-mediated transport. The CAVS risk allele at the lead GWAS SNP at this locus was strongly associated with higher circulating triglycerides, LDL-cholesterol, apolipoprotein B, C-reactive protein, leucocyte and platelet counts, suggesting potential roles in lipids, inflammation and

thrombosis. *FES* encodes a cytoplasmic protein-kinase involved in the regulation of various cellular functions including cell proliferation and differentiation as well as inflammation. This locus has been associated with CAD in several GWAS; the lead CAD SNP, rs17514846, is in high LD ($r^2 > 0.8$) with our lead CAVS SNP. A recent functional study showed that the risk genotype reduces *FES* expression in monocytes and increases the number of monocytes in human atherosclerotic plaque. Knockout of *FES* also promoted migration of monocytes as well as vascular smooth muscle cells and increased atherosclerotic plaque size in a murine model[31]. A decrease in *FES* expression in the aortic valve could lead to CAVS through a similar mechanism. Of note, the risk allele at the lead SNP was also strongly and positively associated with systolic, diastolic and pulse pressure. *AFAP1* encodes actin filament associated protein 1, a potential modulator of actin filament organization in response to cellular signals. AFAP1 was shown to colocalize and bind directly with actin filaments in fibroblast cells. In the same study, deletion of a leucine zipper motif within AFAP1 impacted cell shape and lamellipodia formation in multiple cell lines[32]. This locus was recently associated with descending thoracic aorta diameter by GWAS[33].

Combining multiple lines of evidence to prioritize genes including transcriptomic characterization of a large number of human aortic valves, we highlight a potential role in CAVS for three other genes at novel genome-wide loci: *MUC4*, *ALDH1A2*, and *SMAD9*. *MUC4*, or mucin 4, encodes an integral membrane glycoprotein found on the cell surface. This locus was associated with bicuspid aortic valve in a recent GWAS[34]. In the same study, functional experiments on a zebrafish model showed that knockout or knockdown of *MUC4* leads to a temporal delay in cardiac valve development. In our study, genetically determined expression in human aortic valve was positively associated with CAVS in MR analyses, further implicating this gene in disease pathophysiology. *ALDH1A2*, aldehyde dehydrogenase 1 family member A2, encodes an enzyme that catalyzes the synthesis of retinoic acid from retinaldehyde. Loss-of-function mutations in this gene lead to an autosomal recessive condition presenting with diaphragmatic hernia and cardiovascular defects, including atrial as well as ventricular septal defects and aortic root ectasia[35,36]. In our study, we report a high expression specificity for the aortic valve and a positive relationship between genetically determined expression and CAVS supported by

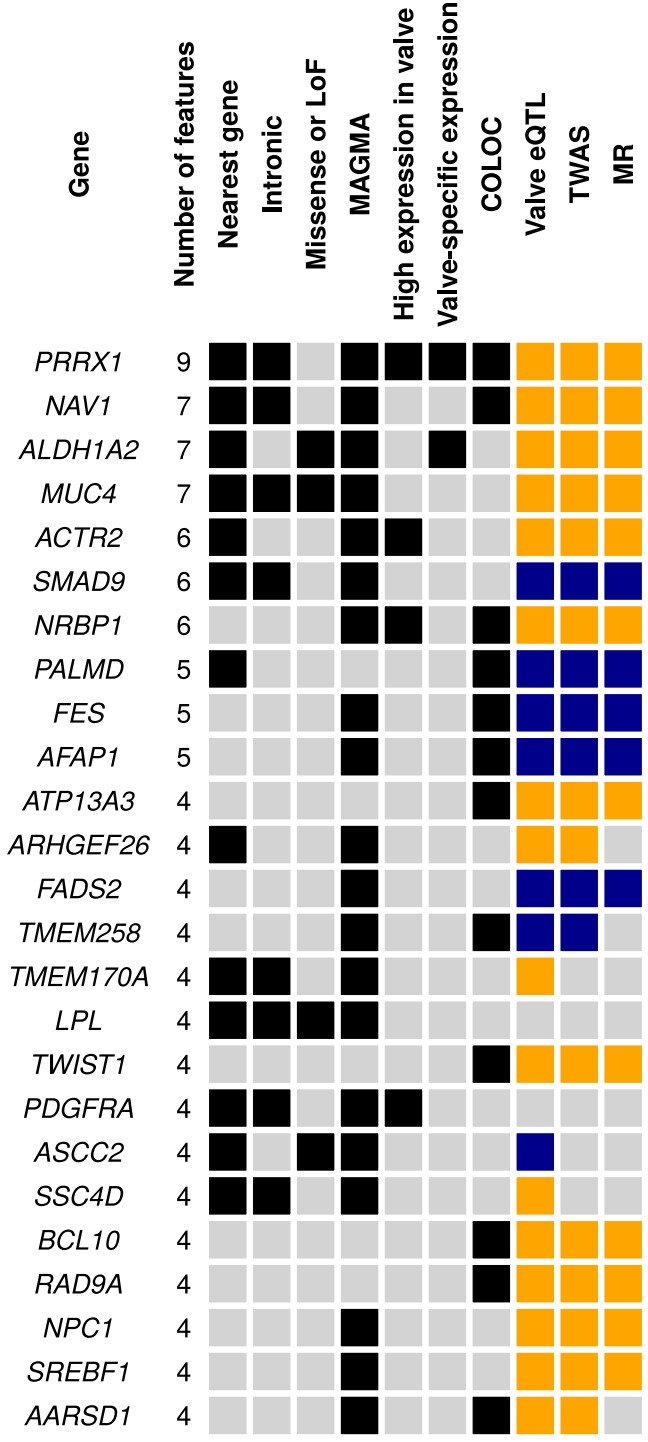

**Fig. 4 | Genes with multiple features suggesting their implication in CAVS.**
Nearest gene: gene closest to a lead SNP in the GWAS meta-analysis; Intronic: annotation of the lead SNP in the meta-analysis; Missense or nonsense: lead GWAS SNP is in linkage disequilibrium ($r^2 \geq 0.8$) with a missense or nonsense variant for the gene; MAGMA: significant in MAGMA analysis at $P < 0.00039$ corresponding to false discovery rate <5%; High expression in valve: above the 90th percentile of all protein-coding genes; Valve-specific expression: expression specificity score >0.1; COLOC: colocalization PP4 > 0.75; Valve eQTL: significant eQTL; TWAS: significant in transcriptome-wide association study at $P < 0.00017$ corresponding to false discovery rate <5%; MR: significant in Mendelian randomization analyses ($P < 0.05$). Gold squares indicate a significant positive association; Blue squares indicate a significant negative association.

MR. Of note, a missense variant in high LD with the lead SNP was inversely associated with CAVS risk. *SMAD9* encodes a member of the SMAD family, which transduces signals from TGF-beta family members. It was shown to inhibit bone morphogenetic protein (BMP) signaling[37]. Loss-of-function mutations in mice models were reported to induce thickening and smooth muscle hyperplasia in the pulmonary vasculature, with upregulation of TGF-beta signaling and *Prx1* (orthologous to human *PRRX1*) expression[38]. A rare missense mutation in humans was associated with unexplained high bone mass[39]. In our study, the lead GWAS SNP was a strong eQTL in the aortic valve. We found a significant inverse relationship between *SMAD9* expression in the aortic valve and CAVS, consistent with the inhibitory effect of this protein on TGF-beta signaling and bone formation.

Several identified loci were associated with circulating blood lipids. While a strong association at the *LPA* locus has previously been reported[7–9,11], the signals in *LDLR* and *LPL* are novel, further implicating atherogenic lipoproteins in the pathophysiology of CAVS. A higher incidence of CAVS has been observed in patients with familial hypercholesterolemia, most frequently caused by a loss-of-function mutation in the *LDLR* gene[40]. The protective allele of the lead variant at the *LPL* locus was in high LD ($r^2 = 0.958$) with rs328 (S447X), a nonsense gain-of-function variant previously shown to increase lipolytic activity[41]. It has also consistently been associated with lower triglycerides levels and lower coronary artery disease risk[42]. Other novel loci associated with blood lipids included 9q34.2 (nearest genes *MYMK-SLC2A6*), for which the lead SNP was associated with circulating LDL-cholesterol and apolipoprotein B and 15q15.2 (nearest gene *STARD9*), for which the lead SNP was associated with plasma triglycerides. Cross phenotype analysis with other GWAS and genetic correlation also showed a significant genetic relationship with circulating lipids, consistent with evidence from previous Mendelian randomization studies[43,44]. All in all, our findings support a significant contribution of circulating blood lipids to the development of CAVS, suggesting a value for lipid-lowering interventions for CAVS prevention.

There were also several loci with a significant association with blood pressure traits, most notably for pulse pressure. In fact, 11 lead SNPs had a significant positive association with pulse pressure in UK Biobank. Among the novel loci, a strong association was observed at 4q12 (nearest gene *PDGFRA*), 11p15.5 (nearest gene *PIDD1*), 12p12.2 (nearest gene *PDE3A*) and 22q12.2 (nearest gene *ASCC2*), which are all loci previously associated with pulse pressure in a recent GWAS meta-analysis[26]. An increase in pulse pressure reflects a reduction in arterial compliance related to increased arterial stiffness or calcification[45]. Of note, missense mutations in *PDE3A* have been associated with an autosomal dominant form of hypertension. Affected individuals manifest with hyperplastic arterial walls resulting from excessive proliferation of vascular smooth muscle cells, which increases peripheral vascular resistance[46,47]. Genetic correlation results showed consistent positive relationships between CAVS and pulse pressure as well as aortic calcification, further supporting shared genetic mechanisms with vascular remodeling.

This study has some limitations. First, the cohorts only included individuals of European ancestry, due to a lack of sufficient available data for other ancestries. Further studies are needed in individuals of other ancestry to determine if the findings are generalizable and to identify specific associations in non-European populations. Second, complementary evidence is needed to understand the biological mechanisms by which the identified genes impact CAVS risk. Third, except for QUEBEC-CAVS-1, the cohorts all included CAVS patients with a bicuspid aortic valve, which could partly mediate the risk for some loci. However, the genes with supportive evidence from the transcriptomic analyses had similar genetic regulation in tricuspid and bicuspid aortic valves. Fourth, the directionality and causal nature of the relationships identified in the cross-phenotype analyses remain to be determined.

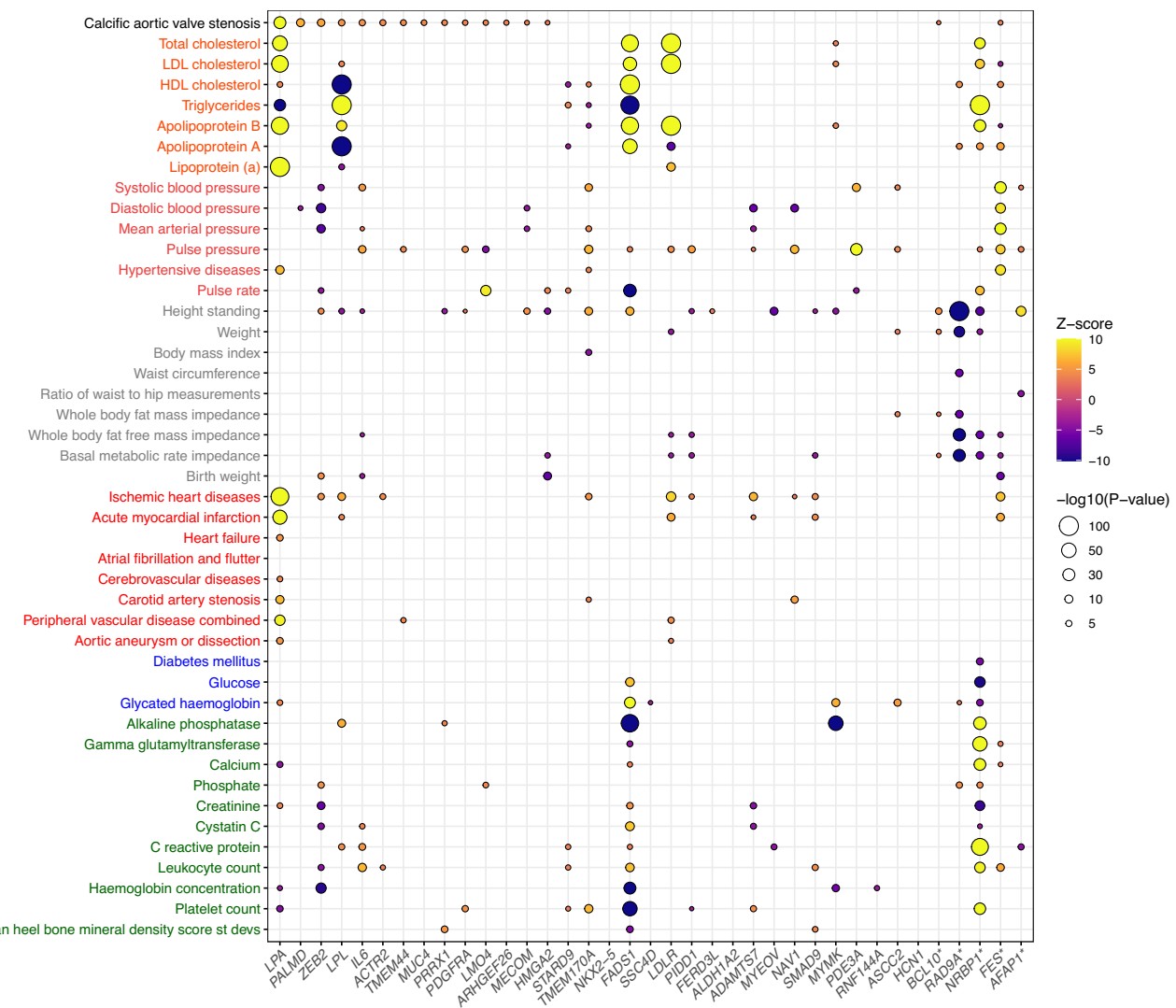

**Fig. 5 | Lead SNPs at GWAS and TWAS loci are associated with cardiovascular traits in UK Biobank.** The nearest gene is indicated on the x-axis; * denotes TWAS loci. The traits are from the following categories: blood lipids (orange), blood pressure (brown), anthropometric traits (gray), cardiovascular diseases (red), diabetes (blue) and others (green). The effect size and statistical significance of the genetic association with the phenotype in UK Biobank are illustrated by the color (Z-score) and the size of the bubble (p-value).

In conclusion, the integration of the largest GWAS meta-analysis to date for CAVS with a large transcriptomic dataset on human aortic valves allowed the identification of novel genomic loci and several candidate causal genes. We highlight the potential contribution of endothelial-mesenchymal transition, circulating lipoproteins, blood pressure, vascular remodeling and inflammation in the disease process. These findings pave the way for the identification of novel therapeutic targets for CAVS.

## Methods

### Ethical approval

The QUEBEC-CAVS study was approved by the ethics committee of the Institut universitaire de cardiologie et de pneumologie de Québec-Université Laval. The Norwich Local Research Ethics Committee granted ethical approval for the analysis in the European Prospective Investigation into Cancer and Nutrition [EPIC]-Norfolk study. The analysis in the Estonian Biobank [EstBB] was approved by the Estonian Committee on Bioethics and Human Research. UK Biobank received approval from the British National Health Service, North West - Haydock Research Ethics Committee. The Coordinating Ethics Committee of the Hospital District of Helsinki and Uusimaa

(HUS) approved the FinnGen study protocol. Informed consent was obtained for all participants.

### Study population

A total of six cohorts were included in the genome-wide association analyses: two case-control cohorts (QUEBEC-CAVS-1 and QUEBEC-CAVS-2) and four population-based cohorts (EPIC-Norfolk, EstBB, UK Biobank and FinnGen) (Supplementary Fig. 1). Only individuals of European ancestry were included, since the sample size was too low for other ancestries. A description of the cohorts including the diagnostic criteria used to define CAVS is available in the Supplementary Information. The analyses in UK Biobank were conducted under data application number 25205.

### Genetic association analyses

Genotyping was completed in each cohort using genome-wide arrays (Supplementary Data 24). Samples with low call rate, sex mismatch with self-report, outlier heterozygosity or ancestry outliers were excluded. Related samples were excluded in the cohorts that did not use a method accounting for sample relatedness. Variants with a low call rate or marked deviations from Hardy–Weinberg equilibrium were

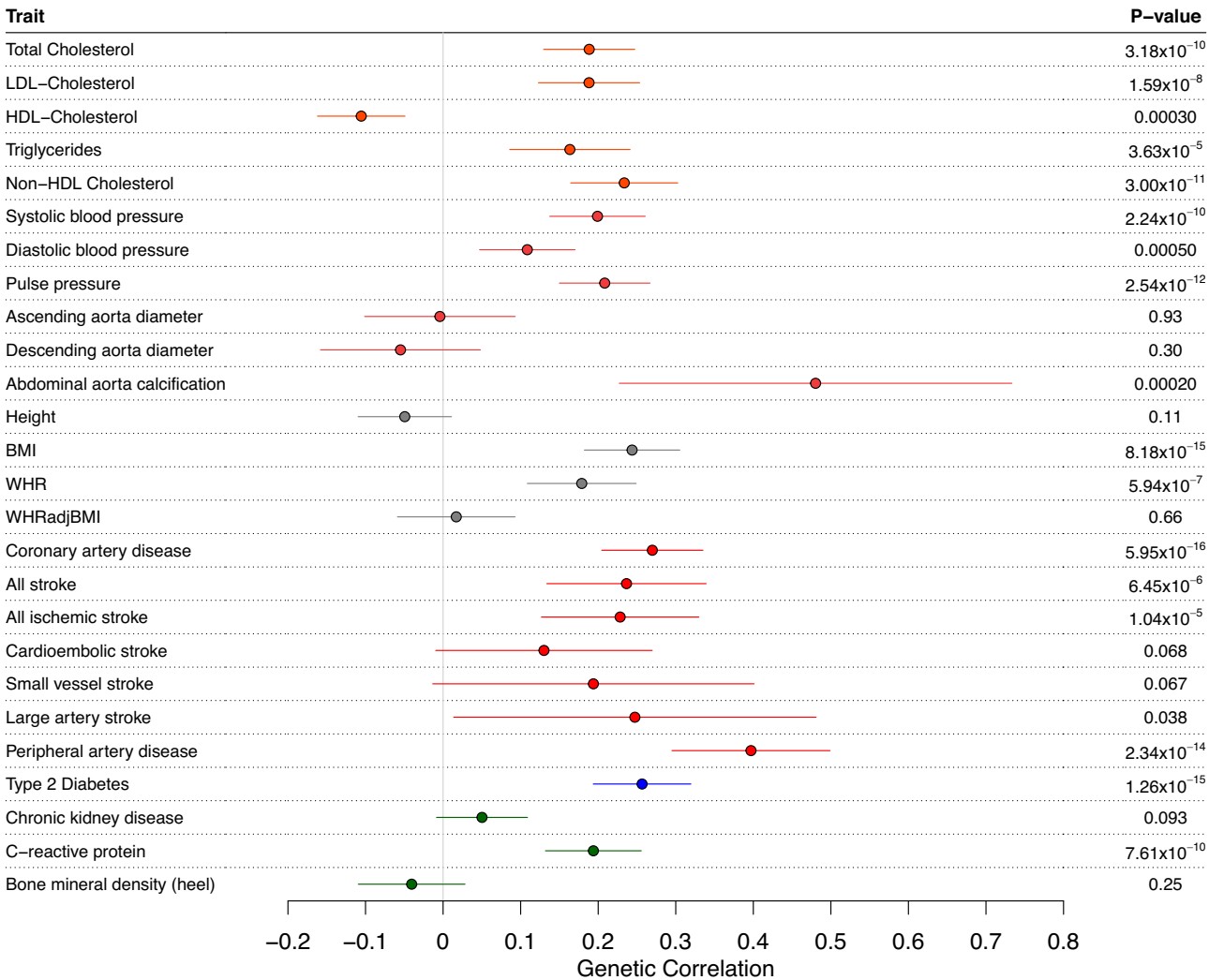

**Fig. 6 | Genetic correlation between CAVS and cardiovascular traits.** Data are presented as genetic correlation (r_g) and 95% confidence interval (+/−1.96*standard error). The traits are from the following categories: blood lipids (orange), blood pressure (brown), anthropometric traits (gray), cardiovascular diseases (red), diabetes (blue) and others (green). LDL low-density lipoprotein, HDL high-density lipoprotein, BMI body mass index, WHR waist-to-hip ratio, WHRadjBMI waist-to-hip ratio adjusted for body mass index.

excluded. Genotypes were imputed using reference panels from Haplotype Reference Consortium, 1000 Genomes, UK10K or a reference panel specific to the population (EstBB and FinnGen) (Supplementary Data 24).

The association between the dosage of each genetic variant and CAVS was evaluated using logistic regression. Covariables included age, sex and ancestry-based principal components (Supplementary Data 24). The summary statistics from each cohort were examined for discrepancies in allele frequencies and inflation factor was calculated. Variants with an imputation quality score <0.3, minor allele frequency <0.001, or minor allele count in cases corrected for the imputation quality <5 (when available) were excluded. An inverse-variance weighted fixed-effect meta-analysis was performed using METAL[48]. The statistical significance threshold was set at $P < 5 \times 10^{-8}$. Heterogeneity was evaluated using Cochran's Q-test. Independent significant variants were identified by applying a linkage disequilibrium (LD) threshold of $r^2 \geq 0.6$ using 1000 Genomes phase 3 European as reference panel to the variants with $P < 5 \times 10^{-8}$. Lead variants were selected by applying a LD threshold of $r^2 \geq 0.1$. Lead variants located within 500 kilobases (kb) were merged into a single genomic locus.

**Replication in other cohorts**

We retrieved the association results for the lead variants from a recent meta-analysis by Chen et al.[14] ($n = 13,765$ cases and 640,102 controls). Although this study also included UK Biobank participants, only up to 1675 cases are common with our analysis (representing ≤11.3% of the cases included in our study). We also retrieved the results from the deCODE cohort[11] ($n = 2457$ cases and 349,342 controls, no overlap with our data). For variants with no results available, we used a proxy in LD ($r^2 > 0.8$) using 1000 Genomes phase 3 European as reference panel.

**Variant and gene prioritization**

Variant annotation was performed using ANNOVAR[49]. Enrichment for functional consequences of variants in LD with independent significant variants ($r^2 \geq 0.6$) was calculated using Fisher's exact test with 1000 Genomes phase 3 European as reference panel. Exonic variants with $P < 1 \times 10^{-5}$ were retrieved to identify missense and loss of function variants in high LD ($r^2 \geq 0.8$) with one of the lead SNPs. A 95% credible set of genetic variants was determined for each genomic locus using CAVIAR, assuming a single causal variant[50]. Conditional and joint association analysis (COJO) was performed using the --cojo-slct

function in GCTA v1.92.3beta3 to identify variants independently associated with CAVS at each genomic locus[51]. Protein-coding genes located within 10 kilobases (kb) of a genome-wide significant SNP or located on either side of a lead intergenic SNP were identified. We also used Multi-marker Analysis of GenoMic Annotation (MAGMA) v1.08 with default settings to perform a gene analysis for 19,982 protein-coding genes[52]. Statistical significance was set below a false discovery rate of 5%.

## Human aortic valve samples

Aortic valve tissues were collected from patients undergoing cardiac surgery at the Institut universitaire de cardiologie et de pneumologie de Québec – Université Laval between 1998 and 2019 (QUEBEC-CAVS-RNA). A total of 500 samples from patients undergoing aortic valve replacement for CAVS ($n = 408$), for aortic regurgitation in the absence of CAVS ($n = 32$) or undergoing heart transplant ($n = 60$, without aortic valve anomaly) were included. Samples were selected to obtain a representation of CAVS with different levels of severity, valve morphology (tricuspid and bicuspid) and patient sex (women and men) (Supplementary Data 10). The presence of aortic valve stenosis and its severity were evaluated using echocardiography, following recognized valve hemodynamic criteria including aortic valve area, mean and peak gradients[53]. Genotypes were obtained from blood samples for each participant using the Illumina Global Screening Array, as described in Supplementary Data 24 for the QUEBEC-CAVS-2 cohort.

## Bulk RNA-sequencing

Aortic valve leaflets were preserved in RNAlater solution following collection and flash frozen in liquid nitrogen. RNA extraction was performed using a modified TRIzol protocol followed by RNA cleanup using QIAGEN RNeasy columns (QIAGEN, 74104). RNA quality was evaluated using a Bioanalyzer. RNA Integrity Number (RIN) was ≥6.0 for all samples selected for sequencing. Libraries were prepared using poly(A) and NEB Library Prep Kit for Illumina to select stranded mRNA (New England Biolabs, E7420L). Expression was assessed by 150 bp length paired-end RNA-sequencing (RNAseq) on a NovaSeq 6000 instrument (Illumina), aiming for >50 million paired reads per sample. The quality of the data was verified using the FastQC v0.11.5 and MultiQC v1.10 applications[54]. STAR v2.5.1b was used to align reads of each sample to the UCSC hg19 and hg38 reference genomes. Alignment quality was verified using QualiMap v2.2.1[55].

## Comparison of expression between aortic valve and other tissues

Reads were collapsed to a single transcript model using GEN-CODE Release 26 on build GRCh38 as reference, to allow direct comparison with the Genotype-Tissue Expression (GTEx) project v8[56]. Read counts and Transcripts Per Million (TPM) values were obtained using RNA-SeQC 2[57]. Genes with a median log2(TPM) value across all samples above the 90th percentile of all protein-coding genes in the aortic valve were identified among the genes prioritized based on their proximity with lead GWAS variants or from the MAGMA analysis. TPM values from 43 non sex-specific tissues in GTEx (Supplementary Data 25) were retrieved. We then calculated expression specificity scores (ESS) for each gene[58]. Briefly, ESS were obtained by dividing the median log2(TPM) value in a given tissue by the sum of the medians of all tissues, resulting in a score ranging from 0 to 1. An ESS above 0.1 in the aortic valve, indicating that expression represents >10% of the overall expression in all tissues, was selected as a threshold to identify genes among the ones prioritized by positional mapping or MAGMA. This threshold corresponded approximately to the top 5% of the distribution of ESS in the aortic valve for all protein-coding genes.

## Expression quantitative trait loci (eQTL) in aortic valve

Further quality control was performed to ensure that the genotypes obtained from blood corresponded to the aortic valve tissue of each participant. Sex mismatches and potential contamination were identified based on *XIST* and *PRKY* expression in the aortic valve RNAseq. The *mbv* function in QTLtools v1.1 was used to match the RNAseq inferred genotypes with the genotypes from blood[59]. Samples with a duplicated sequence, that did not match blood genotype or with potential contamination (>60% of matching heterozygous genotype with another sample) were excluded, which resulted in 484 samples available for further analyses (81.6% with CAVS).

Reads and TPM were generated as mentioned above using GEN-CODE Release 41 on build GRCh37. Genes with expression >0.1 TPM in at least 20% of samples and ≥6 reads in at least 20% of samples were kept[56]. Expression values were normalized between samples using trimmed mean of M-values (TMM) as implemented in edgeR v3.24.3[60]. For each gene, expression was normalized across samples using an inverse normal transformation.

We calculated eQTL for genetic variants located 1 megabase (Mb) up- and downstream of the transcription start site of the genes located on chromosome 1 to 22 (cis-eQTL). Age, sex, smoking status (current or not), the first 60 Probabilistic Estimation of Expression Residuals (PEER)[61] factors and the first five ancestry-based principal components were included as covariates. Only variants with a minor allele frequency ≥0.01 and imputation quality score ≥0.3 were considered. Association was evaluated using QTLtools v1.1[62] with the cis-QTL nominal pass function. We performed cis-QTL permutations using tensorQTL v1.0.7 to generate empirical p-values[63]. Nominal significance thresholds corresponding to a false discovery rate of <5% were calculated for each gene with QTLtools and applied to identify significant eQTLs. A similar analysis was performed separately for tricuspid ($n = 215$) and bicuspid aortic valves ($n = 211$), but using the first 30 PEER factors as covariates considering the smaller sample size.

We retrieved all significant eQTLs for the lead SNPs identified in the CAVS GWAS meta-analysis from our aortic valve dataset. Enrichment for eQTLs was evaluated by comparing the proportion of significant SNP-gene pairs for the lead SNPs to the one for the genome-wide SNPs, using a Pearson's chi-squared test. The distribution of p-values in each group was compared using a Wilcoxon rank sum test. Significant eQTLs for the lead CAVS GWAS meta-analysis SNPs were also retrieved from 43 non sex-specific tissues in GTEx v8. When the lead SNP was not available, a proxy with $r^2 > 0.9$ was used. Tissue-specific eQTLs were defined as significant SNP-gene pairs that were identified in a single tissue.

## Transcriptome-wide association study (TWAS)

Effects of genetic variants on gene expression in the aortic valve were combined with the GWAS meta-analysis results to perform a TWAS using MetaXcan v0.7.4, with the S-PrediXcan extension[64]. First, gene expression models were developed using the PredictDB pipeline[65]. Briefly, elastic-net models were trained using nested cross validation from genotype and adjusted expression data. A model was considered significant when the average Pearson correlation between predicted and observed expression was greater than 0.1 and the estimated *p*-value was less than 0.05. Variants located in a window of +/−1 Mb of the gene of interest were selected. Only protein-coding genes were considered. The statistical significance threshold was set at a false discovery rate <5% to retain genes for further analyses.

## Bayesian colocalization

COLOC v3.2.1 was used to evaluate colocalization between eQTL and CAVS risk for the genes identified by TWAS[66]. Variants located within 1 Mb of the gene were considered. A posterior probability of shared

signal (PP4) > 0.75 was considered as indicative of colocalization. The LocusCompareR package v1.0.0 was used to illustrate colocalization events[67].

## Mendelian randomization (MR)

MR was used to evaluate the causal effect of the changes in gene expression level (exposure) on CAVS (outcome) for the genes identified by TWAS. Variants located within 1 Mb of the gene of interest and associated with gene expression with $P < 0.0001$ were selected. Independent variants were selected using the *clump* function implemented in plink v1.9[68], with a threshold of $r^2 < 0.1$ in the European subset of 1000 Genomes ($n = 503$). Only genes with at least 3 variants available were tested. All the selected variants had an estimated *F*-statistic >15[69]. MR was performed by regressing genetic effect estimates for CAVS as determined in the GWAS meta-analysis on genetic effect estimates for expression levels (inverse-variance weighting). Heterogeneity was evaluated using Cochran's Q statistic, instruments with $P < 0.01$ were not further considered. Egger MR was performed to determine the presence of unmeasured pleiotropy[70]. Weighted median estimator[71] was used to ensure the robustness of the findings. The significance threshold was set at a false discovery rate <5% considering the number of genes tested. The analyses were performed using the *MendelianRandomization* v0.4.3 package.

## Differential expression analyses

To further explore the effect of the risk genotype at a locus that seemed to involve specific regulation in the aortic valve (near *TWIST1*), differential expression analyses were performed. Genes with expression >0.1 TPM in at least 20% of samples and ≥6 reads in at least 20% of samples were selected, resulting in a total of 25,390 genes. Read counts were normalized using TMM for differential expression analyses using edgeR[60]. Samples that did not pass the quality control process (i.e., sex mismatch or mismatch of genotypes with mRNA sequences) were excluded. Among individuals with severe aortic stenosis according to echocardiographic criteria, gene expression in the aortic valve was compared between individuals homozygous for the risk allele rs7804522-C ($n = 66$) and individuals homozygous for the other allele rs7804522-G ($n = 138$) using a likelihood ratio test implemented in edgeR[60]. The model was adjusted for age, sex, smoking, RIN, 3' bias, alignment rate and mitochondrial gene expression proportion. The significance threshold was set at false discovery rate <5%. A differential expression analysis performed using the same method was performed for the other genome-wide significant loci corroborated by transcriptomic analyses in the aortic valve, for comparison purposes.

## Identification of candidate genes from selected features

For the genes located nearby the meta-analysis lead SNPs and the genes identified by TWAS, we counted the number of features that supported their role in CAVS. The following criteria were used: (1) nearest gene to a lead GWAS SNP, (2) lead GWAS SNP is located in an intron of the gene, (3) lead GWAS SNP is in linkage disequilibrium ($r^2 ≥ 0.8$) with a missense or nonsense variant for the gene, (4) gene is significant in the MAGMA analysis, (5) gene has high expression in the aortic valve (above the 90th percentile of all protein-coding genes), (6) gene is specifically expressed in the aortic valve (ESS > 0.1), (7) the lead GWAS SNP is a significant eQTL for the gene in the aortic valve, (8) gene is significant in the TWAS analysis, (9) evidence of colocalization (PP4 > 0.75), (10) evidence of causality between gene expression and CAVS using MR ($P_{IVW} < 0.05$, $P_{WM} < 0.05$ and $P_{Het} > 0.01$).

## Drug target analysis

Among the genes with four or more supporting features, we identified those previously reported to encode a druggable human protein[72]. We then retrieved drugs interacting with these genes (interaction score ≥0.1) from the drug-gene interaction database[73] (DGIdb, www.dgidb.org). A short description of the clinical indication for each drug was retrieved from DrugBank (www.drugbank.ca) or PubChem (https://pubchem.ncbi.nlm.nih.gov/).

## Pathway enrichment

Pathway analyses were performed to identify enriched terms in Gene Ontology biological processes, KEGG pathways, Reactome gene sets, canonical pathways, and WikiPathways using Metascape[74]. Terms with a minimum of five overlapping genes and with a *p*-value < 0.001 obtained from a hypergeometric test are reported. These terms were then hierarchically clustered based on similarities among their gene memberships to identify independent leading terms.

## Phenome-wide association studies

For each lead genetic variant identified from the GWAS or transcriptomic analyses in the aortic valve, association with 44 relevant cardiovascular traits and diseases were evaluated in UK Biobank[9,10]. Phenotypes were obtained from diagnosis in medical records, anthropomorphic measures and laboratory markers (Supplementary Data 26). For each phenotype, an additive logistic (binary phenotype) or linear regression (continuous phenotype) regression was performed in 353,378 unrelated individuals of White-British ancestry with adjustment for age, sex and the first 10 ancestry-based principal components using SNPTEST v2.5.4[75]. The statistical significance threshold was set at $P < 0.001$ considering multiple hypothesis testing.

## Shared genetic association with other traits

The interactive Cross-Phenotype Analysis of GWAS database (iCPAGdb, http://cpag.oit.duke.edu/explore/app/) was used to identify traits with shared genetic association with CAVS[76]. Traits with $P < 0.05$ following Bonferroni correction were considered significantly enriched.

## Genetic correlation

We performed genetic correlation analyses between CAVS and cardiovascular diseases and risk factors using LD-score regression implemented in ldsc v1.0.0[77]. We used publicly available summary statistics from GWAS meta-analyses for blood lipids, blood pressure, aortic size and calcification, height, body mass index, waist-to-hip ratio, coronary artery disease, stroke, peripheral artery disease, type 2 diabetes, chronic kidney disease, C-reactive protein and bone mineral density (Supplementary Data 23).

## Statistical analysis

Analyses were performed using R version 3.5.1 unless otherwise specified.

## Reporting summary

Further information on research design is available in the Nature Portfolio Reporting Summary linked to this article.

# Data availability

Summary statistics of the meta-analysis generated in this study have been deposited in the NHGRI-EBI GWAS catalog under accession code GCST90310293. The RNA sequencing data from the 500 human aortic valves generated in this study have been deposited in dbGaP under accession code phs003541.v1.p1. The data are available under restricted access in accordance with the institutional ethics approval. Access can be obtained for research related to cardiovascular diseases by not-for-profit organizations providing a local institutional review board approval and a letter of collaboration with the study investigators. Requests can be made to the corresponding author who will respond within two weeks. The GWAS summary statistics from FinnGen are available here: https://www.finngen.fi/en/access_results. The summary statistics of the GWAS meta-analysis for CAVS by Chen et al.[14] used in

this study are available here: https://zenodo.org/records/7829401. The GWAS summary statistics for CAVS from the deCODE cohort used in this study are available here: https://www.decode.com/summarydata/. The GTEx project v8 data used in this study are available here: https://gtexportal.org/home/datasets. Individual-level and genotype data from QUEBEC-CAVS-1 and QUEBEC-CAVS-2 are available under restricted access for legal and ethical reasons. Requests can be made to the corresponding author who will respond within two weeks. Access to individual data from CARTaGENE (https://cartagene.qc.ca/), EPIC-Norfolk (https://www.epic-norfolk.org.uk/), Estonian Biobank (https://genomics.ut.ee/en/content/estonian-biobank), UK Biobank (https://www.ukbiobank.ac.uk/) and FinnGen (https://www.finngen.fi/) is available for registered researchers following the respective application process. Further information on data access is available from the study websites. The drug interaction data used in this study are available from the drug-gene interaction database (www.dgidb.org).

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

## Acknowledgements

QUEBEC-CAVS was supported by the Heart and Stroke Foundation of Canada (G-19-0026386), the Canadian Institutes of Health Research (PJT – 153396, PJT – 162344) and the Fonds de recherche du Québec – Santé (278277). Y.B. holds a Canada Research Chair in Genomics of Heart and Lung Diseases. The EPIC-Norfolk study has received funding from the Medical Research Council (MR/N003284/1 MC-UU_12015/1 and MC_UU_00006/1) and Cancer Research UK (C864/A14136). The genetics work in the EPIC-Norfolk study was funded by the Medical Research Council (MC_PC_13048). The work of Estonian Genome Center, Univ. of Tartu was funded by the European Union through Horizon 2020 research and innovation programme under grants no. 810645 and 894987, through the European Regional Development Fund projects GEN-TRANSMED (2014-2020.4.01.15-0012), MOBERA5 (Norface Network project no 462.16.107), MOBEC008, MOBERA21 and Estonian Research Council Grants PUT1660 and PRG1291. We thank the research team at the cardiac surgical database and biobank of the Institut universitaire de cardiologie et de pneumologie de Québec-Université Laval for their valuable assistance and all the participants. We thank the CARTaGENE study team and participants. We are grateful to all the participants who have been part of the EPIC-Norfolk study and to the many members of the study teams at the University of Cambridge who have enabled this research. We want to acknowledge the participants and investigators of

Estonian biobank for their contribution. This work was carried out in part in the High Performance Computing Center of the University of Tartu. We thank all the UK Biobank participants and administration team. We want to acknowledge the participants and investigators of the Finn-Gen study.

## Author contributions

S.T., S.M.B., N.J.W., T.E. and Y.B. supervised the cohort studies. H.D.M., D.K.B., N.G., L.F., D.A., M.D., F.D., M.A.C. and P.P. contributed to data collection. S.T., Z.L., E.A. and J.L. performed the GWAS. S.T. and Z.L. performed the meta-analysis, eQTL, TWAS, colocalization, MR, pheWAS and genetic correlation analyses. S.T., U.H. and P.Z. performed the differential expression analyses. M.B. performed the iCPAG analysis. S.T., B.J.A., P.M. and Y.B. designed the study. S.T. drafted the manuscript. P.M. and Y.B. edited the manuscript. All authors revised the manuscript prior to submission.

## Competing interests

The authors declare no competing interests.

## Additional information

## Estonian Biobank Research Team

**Tõnu Esko**[3]

A full list of members and their affiliations appears in the Supplementary Information.

