## [Peer Review File · Nature Communications]

Integrative genomic analyses identify candidate causal genes for calcific aortic valve stenosis involving tissue-specific regulationReviewer #1 (Remarks to the Author):

Calcific aortic valve disease is an important and increasingly common cardiovascular disease with severe morbidity and mortality if left untreated. Currently, no pharmacotherapies exist; replacement of the aortic valve is the only therapeutic option. This is due in large part to a limited understanding of the mechanisms at play in disease initiation and pathogenesis. 2023 has been a year of substantial progress in the genomic study of calcific aortic valve stenosis (CAVS), with recent reports of CAVS GWAS in 14,451 multi-ancestry cases (Small et al, *Circulation*, 2023) and 13,765 cases of European ancestry (Chen et al, *European Heart Journal*, 2023) which have substantially increased the depth and breadth of our understanding of the genetic landscape of CAVS.

The author group of this current manuscript under review have made important contributions to the field's understanding of CAVS genetics, including a TWAS that identified PALMD as a CAVS susceptibility gene (Theriault et al, *Nat Comm*, 2018), genetic analyses that identified IL6, ALPL, and NAV1 (Theriault et al, *Circ Genom Precis Med*, 2019), and a PWAS that further solidified PALMD as a potential molecular target in CAVS (Li et al, *Commun Biol*, 2020). Here, the authors present what I believe to be the largest CAVS GWAS meta-analysis to date, with 14,819 cases and 927,044 controls in subjects of European ancestry. In addition, they performed transcriptomics on aortic valve tissues from 500 human donors with subsequent eQTL and TWAS analyses. The manuscript is clear and well-written, with notable increases in the size of the GWAS and TWAS cohorts over their prior efforts. However, along with the specific points noted below, the study appears somewhat incremental over those previous studies.

1. A replication cohort does not appear to have been utilized in the GWAS meta-analysis. Recent CAVS GWAS efforts (Small et al, Chen et al; see above) have employed large and robust replication cohorts in their analysis pipelines; indeed the inclusion of replicate cohort analyses has grown to be a standard aspect of modern GWAS. The absence of such a cohort here is a major concern. Please clarify your rationale for this approach and the impact of a lack of replication. Certainly, identifying appropriate replication cohorts grows more difficult with larger meta-analyses; replication against a related phenotype such as aortic valve calcification would be welcome.

2. The focus of GWAS studies on cohorts composed predominantly or exclusively of participants of European descent is broadly recognized as a growing problem that impairs and limits clinical translatability. Small et al. employ a multi-ancestry approach, while Chen et al. attempted cross-ancestry replication of genome-wide significant variants in a small set of African and Latin American participants. The authors do not do so; which would be of significant importance and impact.

3. I strongly concur with the authors statement regarding the limitations presented by their inclusion of cases with bicuspid aortic valves. The studies by Small and Chen attempt to minimize any inadvertent inclusion of bicuspid aortic valves in order to focus on the tricuspid aortic valve CAVS, which is widely believed to be driven by different (non-)genetic drivers than bicuspid stenosis. Several of the novel loci in the GWAS meta-analysis may potentially be driven by the inclusion of bicuspid CAVS cases. For example, NKX2-5 is a transcription factor that plays a key role in cardiac development and is associated with bicuspid incidence in mice and humans. As the authors note in their discussion, MUC4 was recently identified as a GWAS hit for bicuspid aortic valve, and loss of Muc4 in zebrafish induces cardiac defects (Gehlen et al, *Cardiovasc Res*, 2023). Similarly, a SNP modulating GATA4 expression in bicuspid-derived iPSCs regulated by TWIST1 (Huang et al, *ATVB*, 2023), and TWIST1 is a transcription factor with well-described functionality in EndMT and cardiac cushion morphogenesis. I have similar concerns re: their inclusion of bicuspid in the transcriptomics cohort. Are certain GWAS/TWAS hits over-represented in the bicuspid &/or tricuspid groups?

4. In the absence of GWAS replication, the manuscript would benefit from validation of the impact of these hits and their potential downstream signaling mechanisms. For example, does abrogation of TWIST1 in VICs in vitro modulate phenotypic responses (e.g. calcification, myofibroblastogenesis,

inflammation, EMT, etc) and/or transcriptional responses, as it does in vivo? How? Such efforts would dramatically improve the impact and novelty of this study.

5. The methods section does not clearly clarify whether raw counts or TPM-normalized counts were input into edgeR for differential expression analysis (re: TWIST1 risk allele vs other allele); it strongly implies that TPM-normalized counts were used. TPM is inappropriate for edgeR, as it includes a normalization pipeline for library size, etc. As per the edgeR user's guide (section 2.8.6, p.17): "edgeR can work with expected counts as output by RSEM, but raw counts are still preferred." RSEM can generate TPM, though these are different from the expected counts that edge R can (non-optimally) work with. Regardless, edgeR DE analysis is best performed with raw counts. Please clarify and revise this and associated analyses if appropriate. If only TPM are available, please explore the usage of limma-voom or similar.

Reviewer #2 (Remarks to the Author):

The study conducted a genome-wide association study (GWAS) meta-analysis including 14,819 cases among 941,863 participants of European ancestry. It reported 32 genomic loci, among which 20 are novel. RNA sequencing of 500 human aortic valves highlighted an enrichment in expression regulation at these loci and allowed the prioritization of candidate causal genes.

Significance to the field: The study provides new insights into the genetic architecture of calcific aortic valve stenosis (CAVS), a condition for which there is currently no medical therapy. The identification of novel genomic loci and several candidate causal genes could pave the way for the identification of novel therapeutic targets for CAVS.

Comparison to established literature: The study builds upon previous research by using a larger sample size and integrating the largest GWAS meta-analysis to date for CAVS with a large transcriptomic dataset on human aortic valves. This approach allowed the identification of novel genomic loci and several candidate causal genes, highlighting the potential contribution of endothelial-mesenchymal transition, circulating lipoproteins, blood pressure, vascular remodelling, and inflammation in the disease process.

Support for conclusions: The conclusions are supported by the data presented, which includes a large-scale GWAS meta-analysis, RNA sequencing of human aortic valves, and the integration of ten different features to prioritise potential causal genes.

Methodology: The study used a combination of genome-wide association analyses, transcriptome-wide association study, colocalisation and Mendelian randomisation analyses, and pathway enrichment analyses. The study population consisted of six cohorts, including two case-control cohorts and four population-based cohorts. Only individuals of European ancestry were included due to the low sample size for other ancestries.

Reproducibility: The study provides a detailed methodology, including the description of the study population, genetic association analyses, and RNA sequencing methods, which should allow for the work to be reproduced.

Major Issues:

1. Lack of diversity: The study only includes individuals of European ancestry due to the low sample size for other ancestries. This could limit the generalisability of the findings to other ethnic groups. Future studies should aim to include a more diverse study population.

2. Validation of findings: The study identifies novel genomic loci and candidate causal genes, but further research is needed to validate these findings. This could involve in vitro or in vivo experiments to confirm the role of these genes in CAVS.

3. Therapeutic implications: The study identifies potential therapeutic targets for CAVS, but it does not explore these in detail. Future research could focus on investigating the therapeutic potential of these targets, for example through drug discovery or repurposing efforts.

Minor issues:

While the authors mention interesting findings such as the missense variant in high LD with the lead SNP that was inversely associated with CAVS risk in the gene SMAD9, they do not provide specific genomic coordinates for this variant. Providing genomic coordinates of significant variants within the main text of the manuscript would be useful for other researchers who may wish to further investigate them further.

This goes for the other missense variants in high LD with a lead SNP in genes such as ALDH1A2 (rs4646626), MUC4 (rs2246901), CDAN1 (rs4265781), etc.

Overall, the study appears to be well-conducted and provides valuable insights into the genetic architecture of CAVS. The identification of novel genomic loci and candidate causal genes could have significant implications for the development of new therapeutic strategies for this condition. However, further research is needed to validate these findings and explore their potential therapeutic implications.

Reviewer #3 (Remarks to the Author):

This paper integrates a large GWAS meta-analysis for calcific aortic valve stenosis with a transcriptomic dataset from human aortic valves to identify genomic loci candidate causal genes. The authors highlight potential contribution to risk of endothelial-mesenchymal transition, circulating lipids, blood pressure, vascular remodeling and inflammation in CAVS. The paper is very well written and the analyses contribute significantly to the literature on CAVS. In particular, the presentation of a new resource of bulk transcriptomic data from 500 aortic valves will be of great use to the community. This is a great example of how new and publicly available omics data can be integrated in genomics studies to improve interpretation and mechanistic insights. I have a few comments and requests for additional methodological details that I hope will improve the manuscript.

It would be helpful to comment on whether any of the replications reached significance after correcting for the number of look ups performed

It might be more informative to use the TWAS results for the gene mapping in the pathway enrichment analyses. The nearby genes might not have a functional role.

The MR significance threshold should probably be adjusted for the number of tests, rather than 0.05 (though fine, in my opinion, not to correct for the different methods used). Also fine to note how many passed the multiple test correction and how many were nominally sig.

It doesn't make sense to separate the up regulated and down regulated genes for pathway enrichment as the authors have done in the section on Differential expression according to genotype at the TWIST1 locus. Pathways can include both inhibitory and excitatory genes.

Was measured differential expression in the RNAseq data a consideration in prioritization of causal genes?

The cross-phenotype analyses are very interesting, and I am left wondering if there is support for a causal relationship between CAVS and these traits. An MR could help illuminate hypotheses for the mechanisms underlying these relationships.

Can the authors expand on why rs7804522 is not an eQTL in GTEx aortic tissue but is such a strong eQTL in their aortic valve expression analysis? It is a common enough variant, the tissue specificity of the eQTL is remarkable, given that TWIST1 is reasonably well expressed in GTEx aortic artery tissue. Also, TWIST1 seems to be strongly expressed in mammary tissue.

That the cohorts only included individuals of European ancestry and that further studies are needed in individuals of other ancestry to determine if the findings are generalizable is noted as a limitation. But I would argue that beyond assessing whether these findings generalize to non Europeans, it is equally, or even more important to identify new mechanisms of population-specific risk in non Europeans. That is, the purpose of studies in non Europeans is not just to assess the generalizability of findings in Europeans (a very Euro-centric view) but to create robust understanding of risk mechanisms relevant to CAVS in other populations.

Some key details are missing from the online methods:

What software was used for the GWAS analyses?

What thresholds were used for call rate, sex mismatch, heterozygosity, etc? Was a genetic relatedness matrix modeled to control for relatedness within biobanks? How were principal components estimated?

There is no section on how the meta analysis was performed. Was it a fixed effects meta? Random?

What was the range of read counts across the sample in the bulk RNAseq?

Were any RNAseq samples excluded for not meeting quality measures before the comparison of expression tests were performed?

How many genes met inclusion quality control criteria for downstream analyses?

Were peers modeled in the differential expression analyses?

NCOMMS-23-28646-T

Integrative genomic analyses identify novel causal genes for calcific aortic valve stenosis involving tissue-specific regulation

We wish to thank the reviewers for their insightful comments. We updated the manuscript accordingly and believe it is significantly improved. Please find below a point-by-point answer to the reviewers' comments.

Reviewer #1

Calcific aortic valve disease is an important and increasingly common cardiovascular disease with severe morbidity and mortality if left untreated. Currently, no pharmacotherapies exist; replacement of the aortic valve is the only therapeutic option. This is due in large part to a limited understanding of the mechanisms at play in disease initiation and pathogenesis. 2023 has been a year of substantial progress in the genomic study of calcific aortic valve stenosis (CAVS), with recent reports of CAVS GWAS in 14,451 multi-ancestry cases (Small et al, Circulation, 2023) and 13,765 cases of European ancestry (Chen et al, European Heart Journal, 2023) which have substantially increased the depth and breadth of our understanding of the genetic landscape of CAVS.

The author group of this current manuscript under review have made important contributions to the field's understanding of CAVS genetics, including a TWAS that identified PALMD as a CAVS susceptibility gene (Theriault et al, Nat Comm, 2018), genetic analyses that identified IL6, ALPL, and NAV1 (Theriault et al, Circ Genom Precis Med, 2019), and a PWAS that further solidified PALMD as a potential molecular target in CAVS (Li et al, Commun Biol, 2020).

1) Here, the authors present what I believe to be the largest CAVS GWAS meta-analysis to date, with 14,819 cases and 927,044 controls in subjects of European ancestry. In addition, they performed transcriptomics on aortic valve tissues from 500 human donors with subsequent eQTL and TWAS analyses. The manuscript is clear and well-written, with notable increases in the size of the GWAS and TWAS cohorts over their prior efforts. However, along with the specific points noted below, the study appears somewhat incremental over those previous studies.

We thank the reviewer for recognizing the value of our work. We believe this study reports several new findings that constitute a significant step forward in the field of genetics of calcific aortic valve stenosis (CAVS). We report 20 novel genomic loci, including three for which we identify a gene with a causal association between expression in human aortic valve and CAVS. We answered the points highlighted by the reviewer below, including a replication step to further support the findings.

2) A replication cohort does not appear to have been utilized in the GWAS meta-analysis. Recent CAVS GWAS efforts (Small et al, Chen et al; see above) have employed large and robust replication cohorts in their analysis pipelines; indeed the inclusion of replicate cohort analyses has grown to be a standard aspect of modern GWAS. The absence of such a cohort here is a major concern. Please clarify your rationale for this approach and the impact of a lack of replication. Certainly, identifying appropriate replication cohorts grows more difficult with larger meta-analyses; replication against a related phenotype such as aortic valve calcification would be welcome.

We opted to include all available data on CAVS in the meta-analysis to maximize power to discover novel loci. Among the 32 loci described, significant heterogeneity in the effect size between cohorts ($P < 0.05$) was noted for only three, indicating that the majority of the associations were of similar magnitude in the six cohorts included.

Nevertheless, we now provide the association of the 32 lead SNPs (proxied by another variant in linkage disequilibrium when unavailable) with CAVS from two other studies. We compared the results with those of the recent meta-analysis by Chen et al. ($n = 13,765$ cases and $640,102$ controls). Although both studies included UK Biobank participants, only up to $1,675$ cases are common to both analyses (representing $\leq 11.3\%$ of the cases included in our study). All 32 lead SNPs (or proxies) had the same direction of effect, 30 showed nominal association with CAVS ($P < 0.05$) and 29 remained significant when using a threshold of false-discovery rate $< 5\%$. The two lead SNPs that did not replicate are located in the genes *HCN1* and *PIDDI1*. Genes in these two loci were not prioritized by our post-GWAS analyses. In the deCODE cohort ($n = 2,457$ cases and $349,342$ controls, no overlap with our data), 28 out of the 32 lead SNPs had the same direction of effect, of which 10 showed nominal association with CAVS ($P < 0.05$) (see new Supplementary Table 3 below).

We have added this information in the Methods, Results and Discussion sections, respectively:

We retrieved the association results for the lead variants from a recent meta-analysis by Chen et al. ($n = 13,765$ cases and $640,102$ controls). Although this study also included UK Biobank participants, only up to $1,675$ cases are common with our analysis (representing $\leq 11.3\%$ of the cases included in our study). We also retrieved the results from the deCODE cohort ($n = 2,457$ cases and $349,342$ controls, no overlap with our data). For variants with no results available, we used a proxy in LD ($r^2 > 0.8$) using 1000 Genomes phase 3 European as reference panel.

The 32 lead variants at the genome-wide associated loci had a concordant direction of effect in the recent meta-analysis by Chen et al. Thirty showed nominal association with CAVS ($P < 0.05$), of which 29 remained significant when using a threshold of false discovery rate $< 5\%$. In the deCODE cohort ($n = 2,457$ CAVS cases and $349,342$ controls), 28 out of the 32 lead variants had the same direction of effect (Supplementary Table 3).

We performed a GWAS meta-analysis including $14,819$ CAVS cases and identified 32 genome-wide significant loci, 20 of which had not been reported before, including 18 with supportive evidence from another study.

Supplementary Table 3. Association between lead variants at genome-wide significant loci and CAVS in other studies

Locus	Lead SNP	Novel	OR	P Meta	OR Chen ¹	P Chen ¹	P FDR Chen ¹	Dir Chen ¹	OR deCODE ²	P deCODE ²	P FDR deCODE ²	Dir deCODE ²
1	rs2077522	Yes	1.084	3.13E-10	1.064	9.86E-06	2.10E-05	Same	1.089	0.012	0.069	Same
2	rs6702619	No	1.181	3.95E-40	1.157	1.21E-25	1.94E-24	Same	1.230	7.42E-10	2.37E-08	Same
3	rs1016819	No	1.109	1.18E-14	1.106	1.56E-10	8.60E-10	Same	1.066	0.086	0.197	Same
4	rs682112	No	1.108	3.23E-14	1.097	1.61E-10	8.60E-10	Same	1.087	0.023	0.076	Same
5	rs771264	Yes	1.071	4.98E-08	1.050	0.00051	0.00090	Same	1.017	0.609	0.696	Same
6	rs62139061	No	1.090	3.96E-10	1.088	5.53E-08	2.21E-07	Same	1.061	0.105	0.203	Same
7	rs72854462	No	1.152	1.68E-22	1.113	7.92E-11	6.34E-10	Same	1.145	0.00039	0.0031	Same
8	rs111825950	No	1.111	1.06E-10	1.076	6.10E-06	1.39E-05	Same	1.038	0.365	0.510	Same
9	rs11330858	No	1.081	1.31E-09	1.069	1.75E-06	5.09E-06	Same	1.085	0.017	0.069	Same
10	rs1706003	Yes	1.088	3.13E-11	1.044	0.0073	0.0094	Same	1.005	0.871	0.899	Same
11	rs3103933	Yes	1.120	6.12E-09	1.076	1.99E-06	5.30E-06	Same	1.053	0.161	0.286	Same
12	rs147558377	Yes	1.103	2.44E-08	1.075	0.00020	0.00037	Same	0.982	0.728	0.804	Reverse
13	rs182432302	Yes	1.328	2.28E-08	1.152	0.304	0.304	Same	1.151	0.560	0.689	Same
14	rs1965668	Yes	1.081	1.61E-08	1.057	0.0010	0.0016	Same	1.089	0.017	0.069	Same
15	rs10455872	No	1.528	2.97E-61	1.422	4.62E-44	1.48E-42	Same	1.398	1.79E-07	2.87E-06	Same
16	rs7804522	Yes	1.075	1.87E-08	1.051	0.00056	0.00094	Same	1.046	0.189	0.318	Same
17	rs1800797	No	1.121	1.28E-19	1.131	2.90E-18	3.10E-17	Same	1.130	0.00029	0.0031	Same
18	rs2286427	Yes	1.098	4.60E-08	1.045	0.013	0.014	Same	1.084	0.062	0.167	Same
19	rs11570891	Yes	1.205	7.19E-09	1.063	0.0080	0.0094	Same	1.054	0.366	0.510	Same
20	rs17810852	Yes	1.076	1.07E-08	1.054	0.00019	0.00037	Same	1.065	0.063	0.167	Same
21	rs187229435	Yes	1.292	1.60E-09	1.078	0.235	0.243	Same	0.932	0.521	0.667	Reverse
22	rs174551	No	1.089	3.10E-10	1.077	6.41E-07	2.05E-06	Same	0.978	0.505	0.667	Reverse
23	rs12270146	Yes	1.088	8.33E-10	1.047	0.0046	0.0062	Same	1.020	0.601	0.696	Same
24	rs10770612	Yes	1.117	6.56E-13	1.059	0.0012	0.0018	Same	1.078	0.068	0.167	Same
25	rs17766960	Yes	1.105	1.63E-09	1.095	2.95E-07	1.05E-06	Same	1.076	0.108	0.203	Same
26	rs12429277	Yes	1.084	2.31E-08	1.034	0.048	0.051	Same	1.009	0.832	0.887	Same
27	rs28744617	No	1.087	2.25E-08	1.046	0.0078	0.0094	Same	1.042	0.297	0.453	Same
28	rs4646642	Yes	1.077	4.16E-09	1.046	0.0013	0.0018	Same	1.059	0.094	0.200	Same
29	rs2869553	Yes	1.085	2.88E-08	1.043	0.013	0.014	Same	0.996	0.922	0.922	Reverse
30	rs150429885	No	1.083	1.87E-09	1.086	1.41E-08	6.42E-08	Same	1.081	0.024	0.076	Same
31	rs6511720	Yes	1.122	2.50E-08	1.110	3.81E-06	9.38E-06	Same	1.161	0.014	0.069	Same
32	rs61736786	Yes	1.188	3.51E-08	1.136	0.0017	0.0024	Same	1.106	0.268	0.429	Same

1.Chen, H.Y. et al. Dyslipidemia, inflammation, calcification, and adiposity in aortic stenosis: a genome-wide study. Eur Heart J (2023).

2.Helgadottir, A. et al. Genome-wide analysis yields new loci associating with aortic valve stenosis. Nat Commun 9, 987 (2018).

OR: odds ratio for CAVS; P FDR: p-value adjusted for a false-discovery rate of 5%; Dir: Direction of effect

For variants with no results available, we used a proxy in linkage disequilibrium ($r^2 > 0.8$) using 1000 Genomes phase 3 European as reference panel.

3) The focus of GWAS studies on cohorts composed predominantly or exclusively of participants of European descent is broadly recognized as a growing problem that impairs and limits clinical translatability. Small et al. employ a multi-ancestry approach, while Chen et al. attempted cross-ancestry replication of genome-wide significant variants in a small set of African and Latin American participants. The authors do not do so; which would be of significant importance and impact.

We agree that the lack of ethnic diversity is a limitation of the study. CAVS is a disease with a modest prevalence in the population (about 1%), which makes it difficult to gather sufficient power for diverse ethnic groups. We currently do not have access to sufficiently large cohorts of individuals from non-European ancestry to perform a meaningful analysis.

We modified the Discussion to better acknowledge this limitation:

First, the cohorts only included individuals of European ancestry, due to a lack of sufficient available data for other ancestries. Further studies are needed in individuals of other ancestries to determine if the findings are generalizable and to identify specific associations in non-European populations.

4) I strongly concur with the authors statement regarding the limitations presented by their inclusion of cases with bicuspid aortic valves. The studies by Small and Chen attempt to minimize any inadvertent inclusion of bicuspid in order to focus on the tricuspid aortic valve CAVS, which is widely believed to be driven by different (non-)genetic drivers than bicuspid stenosis. Several of the novel loci in the GWAS meta-analysis may potentially be driven by the inclusion of bicuspid CAVS cases. For example, NKX2-5 is a transcription factor that plays a key role in cardiac development and is associated with bicuspid incidence in mice and humans. As the authors note in their discussion, MUC4 was recently identified as a GWAS hit for bicuspid aortic valve, and loss of Muc4 in zebrafish induces cardiac defects (Gehlen et al, Cardiovasc Res, 2023). Similarly, a SNP modulating GATA4 expression in bicuspid-derived iPSCs regulated by TWIST1 (Huang et al, ATVB, 2023), and TWIST1 is a transcription factor with well-described functionality in EndMT and cardiac cushion morphogenesis. I have similar concerns re: their inclusion of bicuspid in the transcriptomics cohort. Are certain GWAS/TWAS hits over-represented in the bicuspid &/or tricuspid groups?

We thank the reviewer for noting this important point. We agree that bicuspid aortic valve (BAV) might have a specific genetic origin and constitutes a significant risk factor for CAVS. However, in most population-based studies, the aortic valve morphology is not assessed systematically and BAV is not recorded specifically. Since we did not have access to valve morphology for the majority of cases included in the meta-analysis, we performed the analysis on all cases of confirmed aortic valve stenosis. A similar approach was used in the previous CAVS GWAS meta-analyses, including the recent studies by Small et al. and Chen et al. We believe that identifying risk loci for CAVS remains relevant, even if some of the loci could be associated with anomalies of aortic valve morphology.

Response to reviewers

As for the transcriptomics cohort, we generated eQTLs specific to each valve morphology (n=215 tricuspid and n=211 bicuspid aortic valves) to determine if the genetic determinants of expression vary. For the 10 genes identified by the transcriptomic analyses (significant in the TWAS, COLOC and MR analyses), the effect of the lead SNP on gene expression was similar in tricuspid and bicuspid aortic valves (all $P < 0.05$ for heterogeneity) (see new Supplementary Table 16 below).

We added the description of this analysis in the Methods section and included this finding in the Results section:

A similar analysis was performed separately for tricuspid (n=215) and bicuspid aortic valves (n=211), but using the first 30 PEER factors as covariates considering the smaller sample size.

To evaluate a potential impact of valve morphology on gene expression regulation for these genes, we calculated eQTL separately for tricuspid and bicuspid aortic valves. For all 10 genes identified by the transcriptomic analyses, the effect of the lead variant on gene expression was similar in tricuspid and bicuspid valves (all $P < 0.05$ for heterogeneity) (Supplementary Table 16).

We also modified the limitation section:

Third, except for QUEBEC-CAVS-1, the cohorts all included CAVS patients with a bicuspid aortic valve, which could partly mediate the risk for some loci. However, the genes with supportive evidence from the transcriptomic analyses had similar genetic regulation in tricuspid and bicuspid aortic valves.

Supplementary Table 16. eQTL according to valve morphology for the ten genes with supporting evidence from transcriptomic analyses

SNP	Gene	Beta All	SE All	P All	Beta TAV	SE TAV	P TAV	Beta BAV	SE BAV	P BAV	P _{Het}
rs6702619	PALMD	-0.720	0.031	7.07E-119	-0.736	0.037	1.71E-50	-0.695	0.037	9.50E-47	0.428
rs682112	NAV1	0.399	0.029	2.26E-42	0.448	0.040	5.78E-23	0.390	0.042	2.49E-17	0.322
rs1016819	PRRX1	0.295	0.036	2.68E-16	0.262	0.057	7.40E-06	0.310	0.053	1.72E-08	0.531
rs1706003	ATP13A3	0.205	0.022	2.40E-20	0.190	0.035	1.15E-07	0.139	0.041	8.99E-04	0.346
rs7804522	TWIST1	0.236	0.025	5.74E-22	0.213	0.039	1.04E-07	0.247	0.039	1.02E-09	0.524
rs35013514	BCL10	0.227	0.034	1.43E-11	0.216	0.048	1.12E-05	0.236	0.048	1.51E-06	0.767
rs35826789	RAD9A	1.068	0.086	9.78E-36	0.857	0.115	2.62E-12	1.034	0.110	1.01E-17	0.267
rs7586601	NRBP1	0.329	0.027	2.55E-34	0.372	0.036	3.53E-20	0.278	0.037	1.10E-12	0.069
rs11372849	FES	-0.265	0.028	9.61E-22	-0.244	0.045	1.54E-07	-0.334	0.037	1.78E-16	0.122
rs4619890	AFAP1	-0.106	0.014	6.64E-14	-0.092	0.023	6.73E-05	-0.140	0.023	4.66E-09	0.136

Beta: normalized effect of the variant risk allele on gene expression; SE: standard error of the normalized effect of the variant on gene expression

All: analysis including all aortic valves (n=484); TAV: analysis including only tricuspid aortic valves (n=215); BAV: analysis including only bicuspid aortic valves (n=211)

P_{Het}: heterogeneity p-value

5) In the absence of GWAS replication, the manuscript would benefit from validation of the impact of these hits and their potential downstream signaling mechanisms. For example, does abrogation of TWIST1 in VICs in vitro modulate phenotypic responses (e.g. calcification, myofibrogenesis, inflammation, EMT, etc) and/or transcriptional responses, as it does in vivo? How? Such efforts would dramatically improve the impact and novelty of this study.

Our transcriptomic analyses on 500 human aortic valves extends the biological insights from GWAS data alone. Using this dataset, we demonstrate that the risk genotype at the *TWIST1* locus has a significant impact on the gene expression signature. We also demonstrate that gene expression of *TWIST1* in human aortic valves is causally related to CAVS risk using Mendelian randomization. We believe that this constitutes significant evidence to support the involvement of this gene in CAVS. Performing in vitro studies would require extensive work to converge to robust conclusions and could therefore be the object of another study. Note that no biological data from aortic valves nor in vitro studies were provided in the two recent GWAS meta-analysis quoted by the reviewer.

6) The methods section does not clearly clarify whether raw counts or TPM-normalized counts were input into edgeR for differential expression analysis (re: TWIST1 risk allele vs other allele); it strongly implies that TPM-normalized counts were used. TPM is inappropriate for edgeR, as it includes a normalization pipeline for library size, etc. As per the edgeR user's guide (section 2.8.6, p.17): "edgeR can work with expected counts as output by RSEM, but raw counts are still preferred." RSEM can generate TPM, though these are different from the expected counts that edge R can (non-optimally) work with. Regardless, edgeR DE analysis is best performed with raw counts. Please clarify and revise this and associated analyses if appropriate. If only TPM are available, please explore the usage of limma-voom or similar.

We indeed used raw counts as input for the edgeR differential expression analyses.

We added this information to the Methods section:

Read counts were normalized using trimmed mean of M-values (TMM) for differential expression analyses using edgeR.

Reviewer #2

The study conducted a genome-wide association study (GWAS) meta-analysis including 14,819 cases among 941,863 participants of European ancestry. It reported 32 genomic loci, among which 20 are novel. RNA sequencing of 500 human aortic valves highlighted an enrichment in expression regulation at these loci and allowed the prioritization of candidate causal genes.

Significance to the field: The study provides new insights into the genetic architecture of calcific aortic valve stenosis (CAVS), a condition for which there is currently no medical therapy. The identification of novel genomic loci and several candidate causal genes could pave the way for the identification of novel therapeutic targets for CAVS.

Comparison to established literature: The study builds upon previous research by using a larger sample size and integrating the largest GWAS meta-analysis to date for CAVS with a large transcriptomic dataset on human aortic valves. This approach allowed the identification of novel genomic loci and several candidate causal genes, highlighting the potential contribution of endothelial-mesenchymal transition, circulating lipoproteins, blood pressure, vascular remodelling, and inflammation in the disease process.

Support for conclusions: The conclusions are supported by the data presented, which includes a large-scale GWAS meta-analysis, RNA sequencing of human aortic valves, and the integration of ten different features to prioritise potential causal genes.

Methodology: The study used a combination of genome-wide association analyses, transcriptome-wide association study, colocalisation and Mendelian randomisation analyses, and pathway enrichment analyses. The study population consisted of six cohorts, including two case-control cohorts and four population-based cohorts. Only individuals of European ancestry were included due to the low sample size for other ancestries.

Reproducibility: The study provides a detailed methodology, including the description of the study population, genetic association analyses, and RNA sequencing methods, which should allow for the work to be reproduced.

Major Issues:

1) Lack of diversity: The study only includes individuals of European ancestry due to the low sample size for other ancestries. This could limit the generalisability of the findings to other ethnic groups. Future studies should aim to include a more diverse study population.

We agree with the reviewer that the lack of ethnic diversity is a limitation.

We modified the Discussion to better acknowledge this limitation:

First, the cohorts only included individuals of European ancestry, due to a lack of sufficient available data for other ancestries. Further studies are needed in individuals of other ancestries to determine if the findings are generalizable and to identify specific associations in non-European populations.

2) Validation of findings: The study identifies novel genomic loci and candidate causal genes, but further research is needed to validate these findings. This could involve in vitro or in vivo experiments to confirm the role of these genes in CAVS.

We believe that the use of transcriptomic data from 500 human aortic valves is a form of in vivo validation for our findings. The differential expression analysis according to the risk genotype at the *TWIST1* locus further supports the implication of this gene in CAVS, considering the impact on the biology in aortic valve tissues. Performing further in vitro or in vivo studies would require extensive work and could be the object of another study.

3) Therapeutic implications: The study identifies potential therapeutic targets for CAVS, but it does not explore these in detail. Future research could focus on investigating the therapeutic potential of these targets, for example through drug discovery or repurposing efforts.

We thank the reviewer for this comment. To further explore the potential of the genes identified for the development of novel therapy for CAVS, we performed a drug target analysis. Among the prioritized genes, seven were considered druggable according to a previously described algorithm (PMID 28356508). We also report the drugs with predicted interactions with these genes in the new Supplementary Table 20.

We added this information in the Methods and Results sections:

Among the genes with four or more supporting features, we identified those previously reported to encode a druggable human protein. We then retrieved drugs interacting with these genes (interaction score ≥ 0.1) from the drug-gene interaction database (DGIdb, www.dgldb.org). A short description of the clinical indication for each drug was retrieved from DrugBank (www.drugbank.ca) or PubChem (<https://pubchem.ncbi.nlm.nih.gov/>).

*Out of the 25 prioritized genes, seven encoded druggable human proteins (*MUC4*, *ALDH1A2*, *NRBPI*, *FES*, *PDGFRA*, *LPL* and *NPC1*). Existing drugs interacting with these genes according to the drug-gene interaction database are reported in Supplementary Table 20.*

We also modified the Discussion section:

Leveraging transcriptomic data obtained from RNA sequencing of 500 human aortic valves, we identified novel candidate causal genes, some of which could constitute therapeutic targets.

Minor issues:

4) While the authors mention interesting findings such as the missense variant in high LD with the lead SNP that was inversely associated with CAVS risk in the gene *SMAD9*, they do not provide specific genomic coordinates for this variant. Providing genomic coordinates of significant variants within the main text of the manuscript would be useful for other researchers who may wish to further investigate them further. This goes for the other

missense variants in high LD with a lead SNP in genes such as ALDH1A2 (rs4646626), MUC4 (rs2246901), CDAN1 (rs4265781), etc.

We report the genomic coordinates for all the missense variants in linkage disequilibrium with the lead SNPs in Supplementary Table 6. We believe reporting the rs identifier in the main text is sufficient, since a query using online tools (such as the dbSNP database) allows to quickly find the genomic coordinates of the variants in different genomic builds.

5) Overall, the study appears to be well-conducted and provides valuable insights into the genetic architecture of CAVS. The identification of novel genomic loci and candidate causal genes could have significant implications for the development of new therapeutic strategies for this condition. However, further research is needed to validate these findings and explore their potential therapeutic implications.

We thank the reviewer for recognizing the value of our work. We acknowledge in the limitations section the need for additional evidence to better characterize the genes that we identified:

Second, complementary evidence is needed to understand the biological mechanisms by which the identified genes impact CAVS risk.

We also added a drug target analysis to better explore the potential therapeutic implications (see response to comment 3) from this reviewer above.

Reviewer #3

1) This paper integrates a large GWAS meta-analysis for calcific aortic valve stenosis with a transcriptomic dataset from human aortic valves to identify genomic loci candidate causal genes. The authors highlight potential contribution to risk of endothelial-mesenchymal transition, circulating lipids, blood pressure, vascular remodeling and inflammation in CAVS. The paper is very well written and the analyses contribute significantly to the literature on CAVS. In particular, the presentation of a new resource of bulk transcriptomic data from 500 aortic valves will be of great use to the community. This is a great example of how new and publicly available omics data can be integrated in genomics studies to improve interpretation and mechanistic insights. I have a few comments and requests for additional methodological details that I hope will improve the manuscript.

We thank the reviewer for recognizing the value of our work, especially the generation of transcriptomic data from 500 human aortic valves. We agree that this constitutes a great asset for the scientific community for the study of CAVS.

2) It would be helpful to comment on whether any of the replications reached significance after correcting for the number of look ups performed

Regarding the replication of loci reported previously, out of the 14 that were not genome-wide significant in our analysis (12 from analysis of European ancestry), seven were associated with CAVS with the same direction of effect at nominal significance ($P < 0.05$), out of which six had a significant association at a threshold of false-discovery rate $< 5\%$.

We added this information in the Results section and modified Supplementary Table 4 accordingly:

Variants at seven other loci previously reported in European populations were nominally associated ($P < 0.05$) with a concordant direction of effect, including six that remained significant when using a threshold of false discovery rate $< 5\%$ (Supplementary Table 4).

3) It might be more informative to use the TWAS results for the gene mapping in the pathway enrichment analyses. The nearby genes might not have a functional role.

We thank the reviewer for this suggestion. We performed a pathway analysis including the genes identified from the MAGMA and TWAS analyses (instead of the nearby genes) (see new Supplementary Fig. 8 below).

We now provide these results in the manuscript:

A pathway analysis was performed with the Metascape tool by including the genes identified using MAGMA as well as the 35 genes significant in the TWAS analysis. Among the top significantly enriched terms, we identified regulation of interleukin-6 production, embryonic development, regulation of osteoblast differentiation, response to growth factor and plasma

lipoprotein assembly, remodeling, and clearance (Supplementary Fig. 8 and Supplementary Table 21).

Supplementary Fig. 8. Pathway enrichment for genes of interest.

The terms were selected from Gene Ontology biological processes, KEGG pathways, Reactome gene sets, canonical pathways, and WikiPathways. The analysis was performed using Metascape with 166 genes retrieved from 173 genes of interest selected based on their significance in the MAGMA and TWAS analyses. The statistical significance of the association for each term and the number of overlapping genes are illustrated by the color (p-value) and the size of the bubble.

4) The MR significance threshold should probably be adjusted for the number of tests, rather than 0.05 (though fine, in my opinion, not to correct for the different methods used). Also fine to note how many passed the multiple test correction and how many were nominally sig.

We thank the reviewer for this suggestion. We now provide the results adjusted for a false-discovery rate of 5%. The 10 genes with supportive evidence from the transcriptomic analyses in human aortic valves remained significant in the MR analyses (false-discovery rate <5%).

We modified the Methods and Results sections, as well as Supplementary Table 15 accordingly:

The significance threshold was set at a false discovery rate <5% considering the number of genes tested.

All genes showed a significant association in MR with the inverse-variance weighted and weighted median approaches (P_{IVW} and P_{WM} adjusted for false discovery rate <5%), suggesting a causal association between gene expression in the aortic valve and CAVS.

5) It doesn't make sense to separate the up regulated and down regulated genes for pathway enrichment as the authors have done in the section on Differential expression according to genotype at the TWIST1 locus. Pathways can include both inhibitory and excitatory genes.

We thank the reviewer for this observation. We agree that the genes included in a pathway can have inhibitory or excitatory effects and that upregulated and downregulated genes could share common pathways. We therefore now provide the pathway enrichment analysis for the complete list of dysregulated genes in Figure 3 (see below). We also modified Supplementary Table 19 accordingly.

Fig. 3. Impact of homozygous risk genotype near TWIST1 on aortic valve gene expression

a, Volcano plot representing the differentially expressed genes between individuals homozygous for rs7804522-C and individuals homozygous for rs7804522-G. Red points represent up-regulated genes (n=148). Blue points represent down-regulated genes (n=361). The top 10 up and down-regulated genes are labelled. **b**, Lead independent enriched terms for the dysregulated genes. The statistical significance of the association for each term and the number of overlapping genes are illustrated by the color (p-value) and the size of the bubble.

6) Was measured differential expression in the RNAseq data a consideration in prioritization of causal genes?

We did not perform systematic differential expression analyses for gene prioritization. Considering the complexity and multiple sensitivity analyses that would be required (definition of CAVS phenotype and severity, valve morphology, etc.), we believe these analyses should be the object of another study.

7) The cross-phenotype analyses are very interesting, and I am left wondering if there is support for a causal relationship between CAVS and these traits. An MR could help illuminate hypotheses for the mechanisms underlying these relationships.

Mendelian randomization analyses have been performed previously and identified several risk factors as being causally associated with CAVS, such as hypertension, obesity and blood lipids (PMID 31290937, 31948645 and 32076698). Considering the large number of phenotypes (35 significantly associated phenotypes after Bonferroni correction) and the possibility of bi-directional relationships, we believe such analyses are outside the scope of the current study.

We added a sentence in the Discussion section to address this limitation:

Fourth, the directionality and causal nature of the relationships identified in the cross-phenotype analyses remain to be determined.

8) Can the authors expand on why rs7804522 is not an eQTL in GTEx aortic tissue but is such a strong eQTL in their aortic valve expression analysis? It is a common enough variant, the tissue specificity of the eQTL is remarkable, given that TWIST1 is reasonably well expressed in GTEx aortic artery tissue. Also, TWIST1 seems to be strongly expressed in mammary tissue.

We thank the reviewer for noting this point. We confirm that rs7804522 is not a significant eQTL in any of the GTEx tissue. The association between this variant and *TWIST1* expression was only found in our human aortic valve dataset, making it an aortic-valve specific eQTL. As noted by the reviewer, this variant is common (minor allele frequency of 42% in the cohorts included in our meta-analysis) and *TWIST1* has high expression in other tissues (including the aorta, adipose tissues and breast), which makes it unlikely that a significant association would have been missed in GTEx. This was also the case for the lead variant at the *PALMD* locus, which was only associated with *PALMD* expression in the aortic valve despite high expression levels in other tissues.

We added a sentence in the Discussion to emphasize this finding:

Moreover, the lead variant at this locus (rs7804522, minor allele frequency of 0.42) was a strong eQTL specific to the aortic valve with no reported association in GTEx, despite high expression

levels in other tissues. Such remarkable eQTL specificity for the aortic valve was also observed at the PALMD locus, as reported previously (PMID 29511167).

9) That the cohorts only included individuals of European ancestry and that further studies are needed in individuals of other ancestry to determine if the findings are generalizable is noted as a limitation. But I would argue that beyond assessing whether these findings generalize to non Europeans, it is equally, or even more important to identify new mechanisms of population-specific risk in non Europeans. That is, the purpose of studies in non Europeans is not just to assess the generalizability of findings in Europeans (a very Euro-centric view) but to create robust understanding of risk mechanisms relevant to CAVS in other populations.

We thank the reviewer for this comment. We agree that including more diversity in GWAS is a priority, not only to determine generalizability, but also to identify population-specific associations.

We rephrased the sentence in the Discussion section:

First, the cohorts only included individuals of European ancestry, due to a lack of sufficient available data for other ancestries. Further studies are needed in individuals of other ancestries to determine if the findings are generalizable and to identify specific associations in non-European populations.

10) Some key details are missing from the online methods: What software was used for the GWAS analyses? What thresholds were used for call rate, sex mismatch, heterozygosity, etc? Was a genetic relatedness matrix modeled to control for relatedness within biobanks? How were principal components estimated?

The software and details of the methodology used to perform the GWAS in each cohort are available in Supplementary Table 24.

We added some details in the text to clarify:

*Genotyping was completed in each cohort using genome-wide arrays (**Supplementary Table 24**). Samples with low call rate, sex mismatch, outlier heterozygosity or ancestry outliers were excluded. Related samples were excluded in the cohorts that did not use a method accounting for sample relatedness. Variants with a low call rate or marked deviations from Hardy-Weinberg equilibrium were excluded.*

11) There is no section on how the meta analysis was performed. Was it a fixed effects meta? Random?

The meta-analysis was performed with the METAL software using an inverse-variant weighted fixed-effect design.

We clarified this in the Methods section:

An inverse-variance weighted fixed-effect meta-analysis was performed using METAL.

12) What was the range of read counts across the sample in the bulk RNAseq?

The total number of paired-end reads per sample ranged from 32.1 to 181.7 million.

We added this information in Supplementary Table 10.

13) Were any RNAseq samples excluded for not meeting quality measures before the comparison of expression tests were performed? How many genes met inclusion quality control criteria for downstream analyses? Were peers modeled in the differential expression analyses?

We first excluded samples that did not pass the quality control process (i.e., sex mismatch or mismatch of genotypes with mRNA sequences). Among the remaining samples, 204 were homozygous for the reference allele (n=138) or the risk allele (n=66) at the *TWIST1* locus (rs7804522) and came from patients with severe aortic valve stenosis. These 204 samples were used in the differential expression analysis. After quality control (minimum TPM and read counts criteria), 25,390 genes were considered in the analysis. We did not include PEER factors as covariables in the differential expression analyses. The analyses were adjusted for age, sex, smoking and technical variables (RIN, 3' bias, alignment rate and mitochondrial gene expression proportion).

We modified the Methods section to clarify:

Genes with expression >0.1 TPM in at least 20% of samples and ≥ 6 reads in at least 20% of samples were selected, resulting in a total of 25,390 genes. Read counts were normalized using TMM for differential expression analyses using edgeR. Samples that did not pass the quality control process (i.e., sex mismatch or mismatch of genotypes with mRNA sequences) were excluded. [...] The model was adjusted for age, sex, smoking, RIN, 3' bias, alignment rate and mitochondrial gene expression proportion.

Reviewer #1 (Remarks to the Author):

I thank the authors for their clear and thorough responses to my initial round of comments. They have adequately addressed the majority of my concerns. I commend them for their efforts to introduce replication experiments and to assess tricuspid vs. bicuspid eQTLs. For a study of this type, I no longer see an absolute need for further in vitro validation of hit molecules now that replication cohorts have been included, though such data would be welcome in the future.

I wish to note that all 3 reviewers raised concerns re: the inclusion of only individuals of European ancestry in the cohorts that were examined here. The authors have added an important discussion of this significant limitation, and I am sympathetic to the reality that there is little else they can do with the study as it currently is structured, as non-European cohorts for this and other phenotypes are extremely limited. This is a broader issue that cannot be solved by these authors alone; however the continued publication of (well-conceived, well-conducted) studies such as this one perpetuates this issue.

Reviewer #2 (Remarks to the Author):

Thank you for your detailed and thoughtful responses. The modification in the Discussion section, highlighting the lack of diversity as a limitation, is appreciated. Your explanation regarding the use of transcriptomic data as a form of in vivo validation is reasonable. Additionally, the inclusion of a drug target analysis provides a direct and valuable link between your research and potential clinical implications. Based on your responses and the subsequent revisions made to the manuscript, I believe it is now well-prepared for publication in Nature Communications.

Reviewer #3 (Remarks to the Author):

My comments have been addressed thanks!